# Canonical and non-canonical integrin-based adhesions dynamically interconvert

Fabian Lukas[1,2], Claudia Matthaeus[3,4], Tania López-Hernández[4], Ines Lahmann[5], Nicole Schultz[1], Martin Lehmann[6], Dmytro Puchkov[6], Jan Pielage[7], Volker Haucke[4,8,9] & Tanja Maritzen[1,2] ✉

Adhesions are critical for anchoring cells in their environment, as signaling platforms and for cell migration. In line with these diverse functions different types of cell-matrix adhesions have been described. Best-studied are the canonical integrin-based focal adhesions. In addition, non-canonical integrin adhesions lacking focal adhesion proteins have been discovered. These include reticular adhesions also known as clathrin plaques or flat clathrin lattices, that are enriched in clathrin and other endocytic proteins, as well as extensive adhesion networks and retraction fibers. How these different adhesion types that share a common integrin backbone are related and whether they can interconvert is unknown. Here, we identify the protein stonin1 as a marker for non-canonical αVβ5 integrin-based adhesions and demonstrate by live cell imaging that canonical and non-canonical adhesions can reciprocally interconvert by the selective exchange of components on a stable αVβ5 integrin scaffold. Hence, non-canonical adhesions can serve as points of origin for the generation of canonical focal adhesions.

Adhesive structures are crucial for anchoring cells within the extracellular matrix, as signaling platforms and as transient attachment points during cell migration. In line with these diverse functions of cellular adhesions, cells generate adhesive structures of different compositions and properties (Fig. S1a)[1].

The best-studied adhesion type are focal adhesions which are generally viewed as the canonical adhesion structure. Focal adhesions typically form right behind membrane protrusions of migrating cells by clustering of heterodimeric adhesion receptors of the integrin family which bind via their extracellular domains to diverse extracellular matrix ligands[2,3]. The initial dot-like adhesions can mature into elongated focal adhesions when the cytosolic integrin tails successfully connect via scaffolding proteins such as paxillin and vinculin to the contractile actomyosin cytoskeleton and experience tension[4–6].

In addition to focal adhesions, in recent years non-canonical cell-matrix adhesions have been recognized to play important roles. These include reticular adhesions (RAs)[1,7], that have also been called clathrin plaques[8–10] or flat clathrin lattices[11–14] [henceforth denoted as RAs/plaques[1]]. Similar to their role in focal adhesions, integrins, especially αVβ5 integrin heterodimers, provide the molecular link between RAs/plaques and the extracellular matrix[7]. Classical focal adhesion proteins like paxillin and vinculin are absent from RAs/plaques, while endocytic

[1]Department for Nanophysiology, RPTU Kaiserslautern-Landau, Paul-Ehrlich-Straße 23, 67663 Kaiserslautern, Germany. [2]Membrane Traffic and Cell Motility Group, Leibniz-Forschungsinstitut für Molekulare Pharmakologie, Robert-Roessle-Straße 10, 13125 Berlin, Germany. [3]Biochemistry and Biophysics Center, National Heart, Lung, and Blood Institute, National Institutes of Health, 50 South Drive, Building 50, Bethesda, MD 20892, USA. [4]Department for Molecular Pharmacology and Cell Biology, Leibniz-Forschungsinstitut für Molekulare Pharmakologie, Robert-Roessle-Straße 10, 13125 Berlin, Germany. [5]Developmental Biology/Signal Transduction Group, Max-Delbrück-Center for Molecular Medicine in the Helmholtz Association (MDC), Robert-Rössle-Straße 10, 13125 Berlin, Germany. [6]Cellular Imaging Facility, Leibniz-Forschungsinstitut für Molekulare Pharmakologie, Robert-Roessle-Straße 10, 13125 Berlin, Germany. [7]Department for Zoology and Neurobiology, RPTU Kaiserslautern-Landau, Erwin-Schrödinger-Straße 13, 67663 Kaiserslautern, Germany. [8]Faculty of Biology, Chemistry, Pharmacy, Freie Universität Berlin, 14195 Berlin, Germany. [9]NeuroCure Cluster of Excellence, Charité Universitätsmedizin Berlin, 10117 Berlin, Germany. ✉e-mail: maritzen@rptu.de

factors such as clathrin and endocytic adaptor proteins are enriched[7]. RAs/plaques were reported to form de novo independently of other integrin-based adhesions[7]. Unlike focal adhesions, RAs/plaques do not need to be connected to actin stress fibers for maturation, although they contain actin and actin regulators[7]. The best-understood function of RAs/plaques so far is to ensure cell attachment during mitosis when focal adhesions are disassembled to allow cell rounding[7]. In addition, they function as plasma membrane-sarcomere attachment sites in muscle cells[15,16], as signaling platforms[9,11,17] and as guidance cues[18].

The group of non-canonical adhesions also comprises retraction fibers, thin elongated tubular extensions of the plasma membrane that are anchored to the substratum via integrins. Retraction fibers frequently break off from migrating cells leaving behind cellular "footprints" providing spatial memory to migrating cells[19–22]. The membrane-enclosed actin cables that anchor mitotic cells to the substratum by connecting to RAs/plaques have also been termed retraction fibers[7], although it is unclear whether these are molecularly identical in nature.

Thus, depending on cell type and culture conditions non-canonical integrin-based adhesions can vary greatly in their shape ranging from diffraction-limited puncta to worm-like structures, rings, thin fibers and micron-sized networks. It is still unclear whether these morphologically distinct structures are identical at the molecular level and whether their life cycle intersects with that of canonical adhesions.

To study the biogenesis and interconversion of different types of adhesions, it is essential to identify unique marker proteins. So far, none of the assigned proteins is exclusive to non-canonical adhesions. αVβ5 integrins are also present in focal adhesions, and the identified endocytic factors also localize to endocytic clathrin-positive structures and vesicles. We had previously reported the characterization of a seemingly endocytic protein called stonin1 which localizes to a specific subset of clathrin-positive structures and accumulates at sites of focal adhesion disassembly[23]. Here, we show that stonin1 accumulates in fact strongly and selectively at non-canonical adhesions instead of being enriched at endocytic structures. Using stonin1 to follow the fate of non-canonical adhesions, we discovered that these adhesions on the one hand can arise by remodeling of focal adhesions and on the other hand can also be converted into focal adhesions revealing a close and dynamic relationship between canonical and non-canonical adhesions.

## Results

### Non-canonical αVβ5 integrin adhesions are characterized by the presence of stonin1

Cellular adhesion to the extracellular matrix is mediated by a range of different integrin-based adhesion complexes. Among the different integrins αVβ5 heterodimers appear especially versatile regarding the adhesive structures they form (Fig. 1a, S1a)[7]. To better understand adhesion diversity and their molecular and dynamic relationship, we started by cataloging αVβ5 integrin-positive structures in C2C12 myoblasts that are known to form non-canonical adhesions[24], which fulfill important functions in muscle cells[16]. As expected, we could prominently detect endogenous αVβ5 integrins in canonical focal adhesions which are characterized by the presence of the typical focal adhesion component paxillin (Fig. 1bi) and show little overlap with the endocytic protein AP2 (Fig. 1ci). In addition, we found endogenous αVβ5 integrin as part of diverse non-canonical matrix adhesions lacking the canonical focal adhesion machinery (Fig. 1b-d, S1a). Based on their morphology these non-canonical adhesions could be divided into punctate or worm-like RAs/plaques (Fig. 1b-dii), large highly interconnected adhesion networks (Fig. 1b-diii) and thin long migratory retraction fibers (Fig. 1b-div). RAs/plaques colocalized excessively with endocytic factors such as AP2 (Fig. 1cii), while the extensive αVβ5 integrin networks (Fig. 1b-diii) forming upon long-term culture (Fig. S2a-f) showed only limited overlap with AP2 (Fig. 1ciii). These networks

were not only present in C2C12 cells, but could be observed also in the human retinal pigment epithelium cell line hTERT-RPE1 (Fig. S2g) and in a range of primary cells including fibroblasts, myoblasts and astrocytes (Fig. S2h). Finally, αVβ5 integrins were prominently observed in retraction fibers which displayed hardly any AP2 co-localization (Fig. 1civ). These data show that αVβ5 integrins mark diverse types of non-canonical adhesions, including RAs/plaques, adhesion networks and retraction fibers which exhibit different degrees of association with endocytic proteins.

So far, the study of non-canonical cell-matrix adhesions has been hampered by the lack of a unique marker since the known components of non-canonical adhesions are either part of focal adhesions (e.g. αVβ5 integrin) or of endocytic structures (e.g. clathrin and endocytic adaptors like AP2). Stonin1, however, a presumptive endocytic adaptor, selectively localized to the entire range of αVβ5 integrin-positive non-canonical adhesions comprising punctate RAs/plaques, large networks and retraction fibers (Fig. 1dii-iv). In fact, while stonin1 colocalized excessively with αVβ5 integrin, its presence was anticorrelated with the focal adhesion protein vinculin (Fig. 1e-g). Intriguingly, vinculin-positive focal adhesions and stonin1-labeled non-canonical adhesions sometimes occurred adjacent to each other on the same large αVβ5 integrin scaffold structure (Fig. 1f). Live cell imaging of genome-engineered C2C12 cells expressing EGFP-tagged stonin1 (for controls showing that the genome-editing did not adversely affect protein levels of critical components, see Fig. S3) further demonstrated that stonin1 is not specifically enriched at short-lived endocytic clathrin-coated pits (Fig. 2a) and confirmed its accumulation at RAs/plaques which can be distinguished by their much longer lifetime[8]. In line with the smaller size of endocytic clathrin-coated pits[11], small clathrin structures also colocalized significantly less with stonin1 than larger ones (Fig. 2b). Hence, stonin1 appears not to accumulate at sites of endocytosis, but prominently localizes to non-canonical adhesions.

The selective localization of stonin1 to non-canonical adhesions was also revealed when comparing conditions that promote non-canonical adhesion formation [e.g. long-term culture (Fig. S2b-f) or collagen coating (Fig. 2c, S4a)] with conditions that prevent their formation [e.g. short-term culture (Fig. S2a) or Matrigel coating (Fig. 2c, S4b)]. In the absence of non-canonical adhesions stonin1 was barely detectable (Fig. 2c-d, S2a, S4b), while clathrin displayed a punctate staining pattern characteristic of clathrin-coated endocytic pits (Fig. 2c). These data confirm that stonin1 does not accumulate at endocytic structures, but instead marks non-canonical adhesions. We probed this also genetically by depleting cells of β5 integrin, a component required for the formation of non-canonical adhesions. Loss of β5 integrin resulted in a diffuse cytoplasmic staining for stonin1 (Fig. 2e-g). In contrast, depletion of endocytic structures via deletion of the endocytic adaptor AP2[25] did not affect the localization of stonin1 to integrin-based structures in primary astrocytes from tamoxifen-inducible AP2 knockout mice (Fig. S5). Thus, we conclude that αVβ5 integrins, but not endocytic factors recruit stonin1 to non-canonical adhesions.

### αVβ5/stonin1-positive networks are distinct from RAs/plaques

Given the profound morphological difference between the small punctate non-canonical αVβ5 structures that have been described as RAs/plaques and the much larger αVβ5 networks observed in long-term cultures (Fig. 1b-diii, S2d) or on collagen (Fig. S4a), we wondered to which extent these structures share a similar molecular composition. Since RAs/plaques have been defined by the absence of canonical focal adhesion proteins and the presence of endocytic factors, we performed a detailed localization analysis of endocytic markers within the adhesion networks to determine whether endocytic proteins colocalize with both RAs/plaques and adhesion networks. To avoid antibody species incompatibilities, we used our genome-edited EGFP-

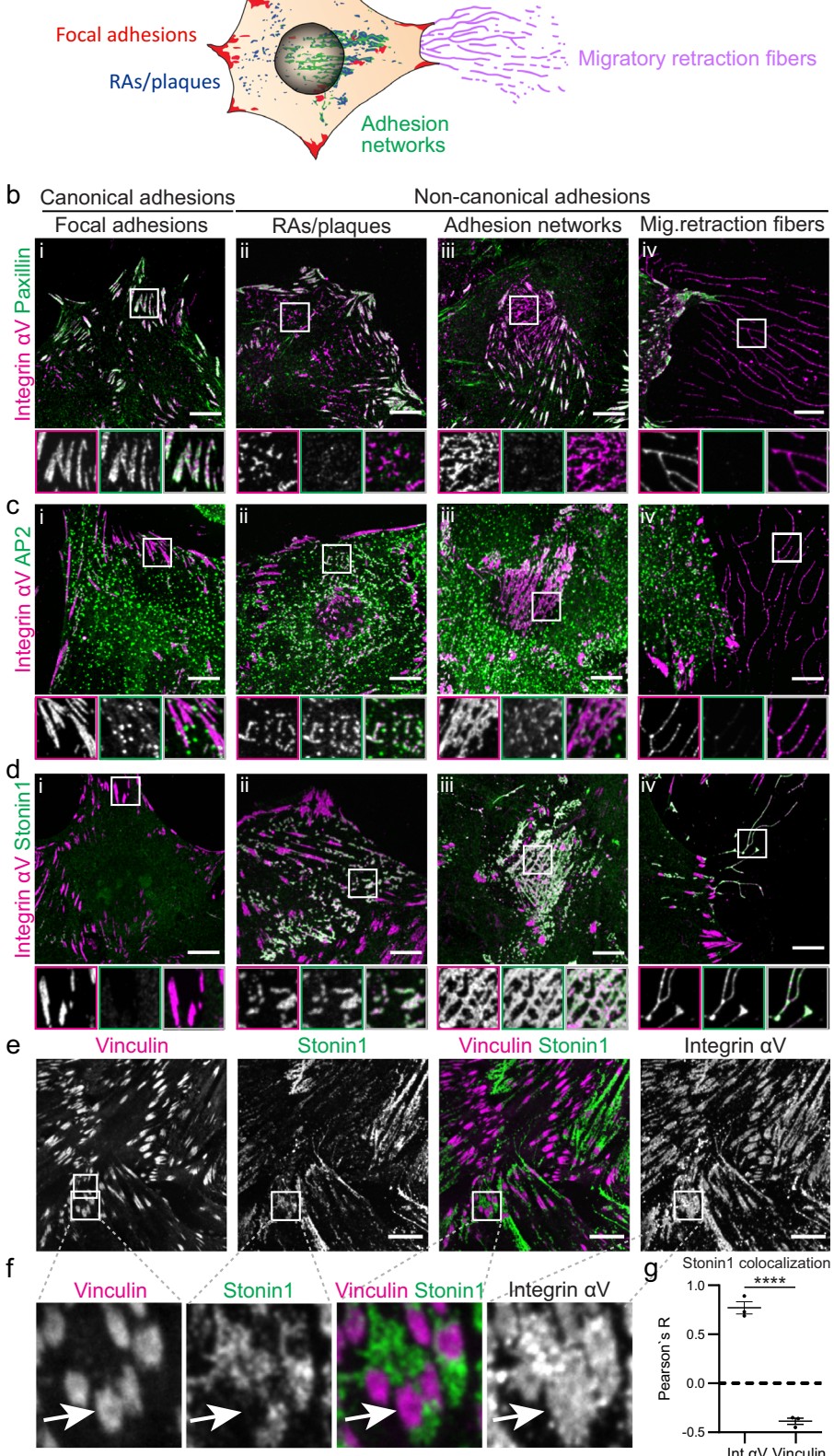

tagged stonin1 as marker for the αVβ5-based networks (Fig. 1diii, 3a). We found that the overlap between endocytic proteins (clathrin, Eps15R, stonin2 and ITSN1) and stonin1 (as proxy for αVβ5 integrin) within networks was very limited (Fig. 3a,b). In all cases, co-localization was confined to interspersed puncta, while large stonin1-positive areas were devoid of endocytic factors.

To be able to visualize also endogenous β5 integrin in the absence of reliable antibodies against the murine protein, we generated a double genome-edited C2C12 cell line expressing EGFP-stonin1 and integrin β5-mScarlet3 from their respective endogenous locus (Fig. S3). β5-mScarlet3 colocalized perfectly with antibody-labeled αV integrin and with EGFP-stonin1 within

**Fig. 1 | Non-canonical αVβ5 adhesions are characterized by the presence of stonin1. a** Illustration of the different types of αVβ5 integrin-positive canonical and non-canonical adhesion structures in a migrating mesenchymal cell. **b–d** C2C12 myoblasts, cultured for different times on vitronectin (i: 24 h, ii: 96 h, iii: 10 days, iv: 96 h) to promote the biogenesis of diverse types of adhesions (i-iv), were immunolabeled for αV integrin in combination with paxillin (**b**) as canonical focal adhesion marker, AP2 (**c**) as RA/plaque component and stonin1 (**d**) as marker for the entire range of paxillin-negative αV integrin-positive adhesions. Scale bar, 10 μm. Insets are shown enlarged as split colour channels (depicted in white) and merged view. RAs, reticular adhesions; mig., migratory. **e–g** C2C12 cells, cultured on collagen for 72 h, were immunolabeled for αV integrin in combination with vinculin as canonical focal adhesion marker and stonin1 to show anti-correlation of vinculin- and stonin1-positive αV integrin structures. Scale bars, 10 μm. **f** Magnification of insets from (**e**). **g** Analysis of stonin1 colocalization with αV integrin and vinculin based on Pearson´s correlation coefficient R. Negative values denote anti-correlation (mean±SEM; $N = 3$ independent experiments, two-tailed unpaired Student´s t-test, ****$p < 0.0001$). Source data are provided as a Source Data file.

large adhesion networks, confirming the correct localization of the tagged protein (Fig. S6). Again, clathrin labeled only punctate structures within the adhesion networks (Fig. S6). This observation held also true in the human hTERT-RPE-1 cell line where we labeled αV and β5 integrin, clathrin and stonin1 with specific antibodies (Fig. S2g). These data suggest that clathrin lattices do not present an essential scaffold for the maintenance of αVβ5/stonin1-positive adhesion networks.

To monitor the dynamics of clathrin in comparison to the dynamics of αVβ5 integrin and stonin1 within adhesion networks, we performed TIRF multicolor live cell imaging in cells expressing genome-edited EGFP-stonin1 and transduced with β5 integrin-iRFP and mRFP-clathrin light chain. β5 integrin and stonin1 were highly stable within adhesion networks and colocalized over extended periods of time (>3 h). In contrast, clathrin puncta that co-localized with parts of the adhesion network appeared and disappeared frequently during the imaging period (Fig. 3c,d).

Earlier studies raised the question whether non-canonical adhesions anchoring cells during mitosis are associated with clathrin[7]. Using multi-color live cell imaging we observed that clathrin remained at αVβ5 adhesions residing within the small cell area that stayed attached upon mitotic cell rounding. However, clathrin completely dissociated from RAs/plaques located outside of this small cell area which represent adhesion sites for mitotic retraction fibers[7] that emanate from the cell body and provide additional anchorage for the rounded cell (Fig. 3e,f; S7; Movie 1). Only when the daughter cells spread back down over these adhesions, we observed clathrin to re-associate with the integrin adhesion scaffold (Fig. 3e,f; S7; Movie 1) underlining the dynamic nature of clathrin at diverse non-canonical adhesions in contrast to the stable presence of αVβ5 and stonin1.

Our finding that non-canonical αVβ5 integrin-positive adhesion networks largely lack clathrin argues for the existence of non-canonical αVβ5 adhesions that are maintained independently of clathrin-scaffolding. To determine if αVβ5/stonin1-positive networks associate with flat clathrin lattices, we applied correlative fluorescence and platinum replica transmission electron microscopy (PREM) to C2C12 cells expressing genome-edited EGFP-stonin1 and β5 integrin-iRFP. Stonin1 and β5 integrin co-localized in large networks at the plasma membrane (Fig. 4a). TEM analysis of platinum replicas revealed that 82% of the stonin1-positive area was devoid of clathrin structures (Fig. 4b,e; S8a). In the rare cases in which clathrin structures were detected within αVβ5/stonin1 adhesions (Fig. 4b,f; S8b), these were flat clathrin lattices (98.5%) rather than clathrin-coated pits or domes (0.3% resp. 1.2%), suggesting that αVβ5-positive clathrin plaques are occasionally interspersed within the larger αVβ5 networks (Fig. 4c,d; S1b). Collectively, these data show that αVβ5/stonin1-positive networks are distinct from clathrin plaques. Nevertheless, the determinants for the formation and maintenance of the large αVβ5/stonin1-positive networks display overlap with criteria established for RAs/plaques[7,13,14,26]. The adhesion networks persist upon loss of tension by treatment with the myosin II inhibitor blebbistatin (Fig. S9a), and like RAs/plaques they represent the preferential location for a β5 integrin mutant mimicking serine phosphorylation of its SERS motif (Fig. S9b).

## Biogenesis of non-canonical αVβ5 integrin adhesions

We know little about the dynamic relationship between different types of αVβ5 adhesions. The fate of focal adhesions and RAs/plaques was suggested to be mainly connected via their shared use of αVβ5 integrins. Manipulations that inhibited one of these structures appeared to favor the other[1,7], presumably by enlarging the available αVβ5 integrin pool. However, it remained unclear whether and how αVβ5 integrin physically transfers between focal adhesions and RAs/plaques, especially since RAs/plaques were shown to form de novo[7].

We decided to revisit the question of RA/plaque biogenesis by capitalizing on stonin1 as a marker for non-canonical αVβ5 integrin adhesions. Live cell imaging of genome-edited EGFP-stonin1 in fact revealed multiple biogenesis modes for non-canonical αVβ5 integrin adhesions. First, our experiments in migrating cells confirmed that RAs/plaques can form de novo, although such events appeared to be rare (Fig. S10a-g). When migrating cells covered a new area, we occasionally observed the simultaneous deposition of β5 integrin and stonin1 after the protrusion of the leading edge. When probing for clathrin in the process of de novo RA/plaque formation, we found that clathrin is co-recruited with β5 integrin and stonin1 when RAs/plaques form de novo (Fig. S10d-g). The initially small adhesion structures could then merge into larger networks over time (Fig. S10h,i). However, more frequently non-canonical adhesions arose at sites where the canonical focal adhesion marker paxillin was progressively replaced by stonin1 (Fig. 5a,b), indicating the successive conversion of focal adhesions into RAs/plaques. In comparison to stonin1, clathrin was recruited to a lesser extent. Throughout the transition from focal adhesions to RAs/plaques, the size, shape and fluorescence intensity of β5 integrin remained unaltered, suggesting that the integrin scaffold was retained throughout the conversion process. We challenged these unexpected findings by live TIRF microscopy to quantify the fluorescence profile of β5 integrin, paxillin and stonin1 over time. While β5 integrin levels remained stable over the entire course of 105 min, paxillin fluorescence was lost, whereas stonin1 levels simultaneously increased over time (Fig. 5c,d; Movie 2). These data were also confirmed when using endogenously tagged β5 integrin instead of overexpressed β5 integrin (Fig. S11a, b).

We also observed the conversion of focal adhesions into migratory retraction fibers (Fig. 5e,f). When migrating cells retracted their cell membrane, focal adhesions often exhibited a "sliding" behavior involving disassembly at the peripheral end and elongation at the internal end. We found that while paxillin was lost from the distal end of the disassembling focal adhesion, the αVβ5 integrin scaffold actually remained intact and became progressively associated with stonin1 thereby turning into a retraction fiber (Fig. 5e,f). The progressively elongating retraction fiber stayed connected with the intracellular actin cytoskeleton, although the thickness of the associated stress fiber bundles decreased over time (Fig. 5f). We conclude that focal adhesions can be successively converted into RAs/plaques or retraction fibers marked by αVβ5 integrin and stonin1.

## Biogenesis of focal adhesions from non-canonical αVβ5 integrin adhesions

Our observations of migrating cells revealed that conversions in the opposite direction also occur. Here, focal adhesions are assembled on

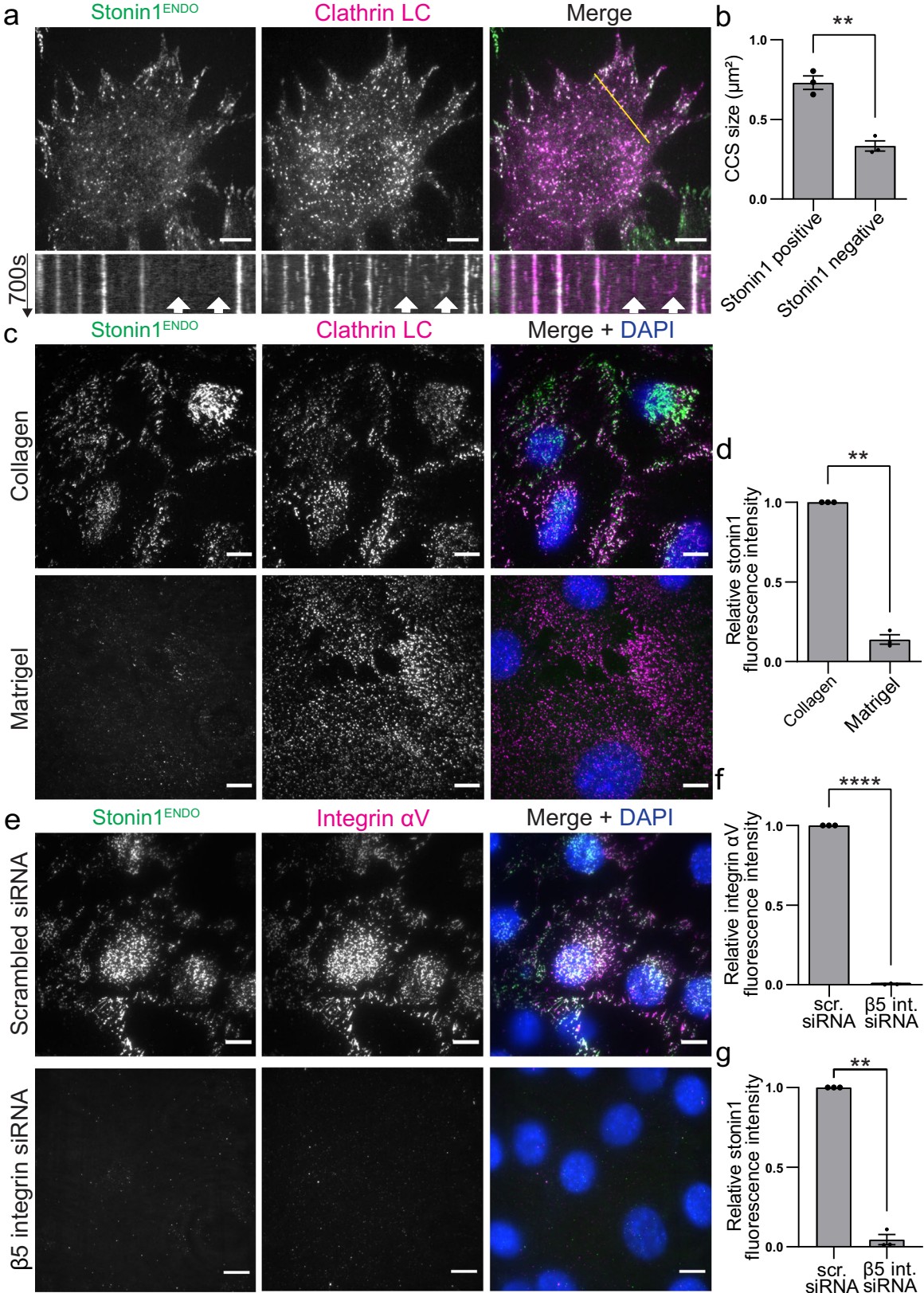

integrin scaffolds derived from pre-existing non-canonical adhesions (Fig. 6a). For example, upon respreading of a migratory cell across a β5 integrin scaffold located within a migratory retraction fiber which the cell previously formed and which is still connected to its cell body, we observed the deposition of the focal adhesion marker paxillin onto the pre-existing paxillin-negative β5 integrin structure such that both proteins extensively overlapped in the end (Fig. 6b; Movie 3). Our quantification of fluorescence profiles of β5 integrin and paxillin across different respreading events from independent experiments confirmed the recruitment of paxillin onto pre-existing β5 integrin scaffolds (Fig. 6c). We further confirmed these data using endogenously tagged β5 integrin instead of overexpressed β5 integrin (Fig. S11c-e).

**Fig. 2 | Stonin1 localizes in integrin αVβ5-dependent manner to non-canonical adhesions while being absent from endocytic structures. a** Stonin1 accumulates with clathrin in long-lived clathrin plaques, but not in short-lived endocytic clathrin-coated pits. C2C12 cells expressing endogenously tagged EGFP-stonin1 and stably transduced with mRFP-clathrin light chain (LC) were imaged live. The yellow line depicts the position chosen for the kymographs depicted below the still images. Scale bar, 10 µm. **b** Quantification of the respective sizes of stonin1-positive and stonin1-negative clathrin-coated structures (CCS) reveals predominant presence of stonin1 in larger clathrin structures i.e. RAs/plaques (mean±SEM, $N = 3$ independent experiments, two-tailed unpaired Student´s t-test, $**p = 0.0017$). **c** Stonin1 localization depends on the presence of non-canonical integrin adhesions rather than clathrin-coated pits. C2C12 cells expressing endogenously tagged EGFP-stonin1 and co-transfected with mRFP-clathrin LC were seeded for 24 h on collagen which promotes non-canonical adhesion formation or Matrigel which suppresses their formation. Even though clathrin-coated pits remain on Matrigel, stonin1 localization is lost. Scale bar, 10 µm. **d** Quantification of stonin1 fluorescence intensity in cells on Matrigel relative to its level in cells on collagen (mean±SEM, $N = 3$ independent experiments, One sample t-test, $**p = 0.0012$). **e** Stonin1 localization depends on the presence of αVβ5 integrin. C2C12 cells expressing endogenously tagged EGFP-stonin1 and treated with β5 integrin-specific or scrambled control siRNA were seeded on collagen for 24 h. Fixed cells were immunolabeled for αV integrin and EGFP. Nuclei were stained with DAPI. Scale bar, 10 µm. **f, g** Quantification of αV integrin (**f**) and stonin1 (**g**) fluorescence intensity in β5 integrin depleted (β5 int siRNA) cells relative to control (scr siRNA) cells (mean ±SEM, N=3 independent experiments, One sample t-test, $**p = 0.0011$, $***p < 0.0001$). Source data are provided as a Source Data file.

This type of conversion also occurred when daughter cells respread after mitosis. Once they reached the sites of RAs/plaques that had provided anchorage for mitotic retraction fibers during cell rounding, the αVβ5 integrin scaffolds of these non-canonical adhesions became decorated with paxillin converting them into focal adhesions (Fig. 6d,e; Movie 4). These observations were subsequently confirmed by using cells expressing endogenously tagged β5 integrin (Fig. S11f-g).

Our results suggest that focal adhesions can be generated on preexisting αVβ5 integrin scaffolds that originally were part of migratory retraction fibers or RAs/plaques. This conversion mechanism likely enables efficient respreading of cells over a previously occupied area since the reallocation of existing αVβ5 integrin scaffolds circumvents the need to cluster integrins delivered from internal pools or from the cell surface.

Given that the conversion mechanisms described thus far operate in migratory and respreading cells, we next asked whether adhesion conversion from non-canonical to canonical adhesions also occurs in confluent stationary cells. To address this, we synchronized focal adhesion assembly by triggering their disassembly via the myosin II inhibitor blebbistatin (Fig. 6f,g, S9a) (a condition leaving the non-canonical adhesion networks intact) and promoting their subsequent reformation by its washout. In the absence of αVβ5 integrin/stonin1-positive networks (i.e. on Matrigel), new focal adhesions formed predominantly at the cell periphery (Fig. 6g,h). However, in the presence of non-canonical adhesion networks (i.e. on collagen), small focal adhesions appeared also centrally right next to the networks (Fig. 6g,h) suggesting again an additional mode of focal adhesion assembly that rather depends on pre-existing αVβ5 integrin scaffolds than on lamellipodial protrusion. However, the molecular details behind this mechanism remain elusive at the moment.

Our experiments point to a long-lived αVβ5 integrin scaffold structure which depending on cellular signals can serve as the backbone for either focal adhesions or non-canonical adhesions and remains largely unchanged when these adhesion types interconvert by the loss and recruitment of accessory proteins to the integrin scaffold. To investigate whether the overall longevity of the integrin αVβ5 scaffold is reflected by a low integrin turnover rate within adhesions, we performed fluorescence recovery after photobleaching (FRAP) experiments with cells expressing endogenously-tagged EGFP-stonin1 and integrin β5-mScarlet3 and overexpressing paxillin-iRFP. Indeed, we observed a very low fluorescence recovery after photobleaching for integrin β5-mScarlet3, irrespective of whether it was residing in focal adhesions or non-canonical adhesions (Fig. 7a,b). This suggests a very large immobile fraction of the molecule, consistent with providing a long-term scaffold structure. In comparison, EGFP-stonin1 as an exchangeable component displayed a much higher fluorescence recovery and thus a higher mobile pool within non-canonical adhesions (Fig. 7a,c).

## Discussion

Our findings reveal that the different αVβ5 integrin-based adhesions, while appearing in diverse shapes and having distinct molecular compositions, are intimately linked and dynamically interconvertible. More specifically, using stonin1 as a bona fide marker of non-canonical adhesions, we demonstrate that (i) focal adhesions can be converted into non-canonical adhesions (i.e. RAs/plaques and migratory retraction fibers). (ii) Conversely, these non-canonical adhesions can be turned into focal adhesions for efficient respreading. (iii) Non-canonical αVβ5 integrin adhesion networks can serve as starting points for the generation of internal focal adhesions in confluent cells.

With these observations, we complement the well-researched mechanism of focal adhesion biogenesis, that is based on the lamellipodial formation and subsequent maturation of nascent adhesions[4], with an additional pathway where existing non-canonical αVβ5 integrin structures can be recycled for the generation of focal adhesions, bypassing the need for gradual integrin clustering and enlarging the options for focal adhesion assembly in different cellular contexts.

During the conversion process, the αVβ5 integrin structure remains essentially unaltered while its interactome changes, constituting a stable scaffold within the dynamically adapting adhesion landscape. For example, in the conversion of focal adhesions to RAs/plaques the canonical focal adhesion marker paxillin is gradually lost from the αVβ5 integrin scaffold while stonin1 and endocytic proteins are simultaneously recruited (Fig. 5a,b). The same happens when focal adhesions turn into migratory retraction fibers. The αVβ5 integrin scaffold at the distal end of the inward sliding focal adhesion is not disassembled, but instead exchanges paxillin for stonin1 thereby turning into a migratory retraction fiber (Fig. 5c,d).

In the opposite direction, αVβ5 integrin scaffolds remaining behind within retraction fibers can re-associate with paxillin upon the respreading of a migrating cell towards its previous position (Fig. 6a-c). Similarly, the αVβ5 integrin-positive non-canonical adhesions attaching mitotic cells to the substrate can acquire paxillin upon respreading of the daughter cells (Fig. 6e,e) allowing the efficient reestablishment of focal adhesions.

Based on these observations we propose a model where a stable, but versatile integrin scaffold structure can either be converted into a focal adhesion by the reciprocal loss of stonin1 and recruitment of canonical focal adhesion proteins or can be turned into an RA/plaque by the association of endocytic proteins (Fig. S1c). Consequently, focal adhesions and RAs/plaques can interconvert via an intermediate αVβ5 integrin- and stonin1-positive adhesion structure. Consequently, our results suggest that the different types of integrin-based adhesions should not be viewed as discrete entities that form and disassemble independently of each other, but rather as an interconvertible and dynamic continuum of adhesion structures that can adapt to the changing (extra-)cellular environment by the selective loss and/or recruitment of molecular components.

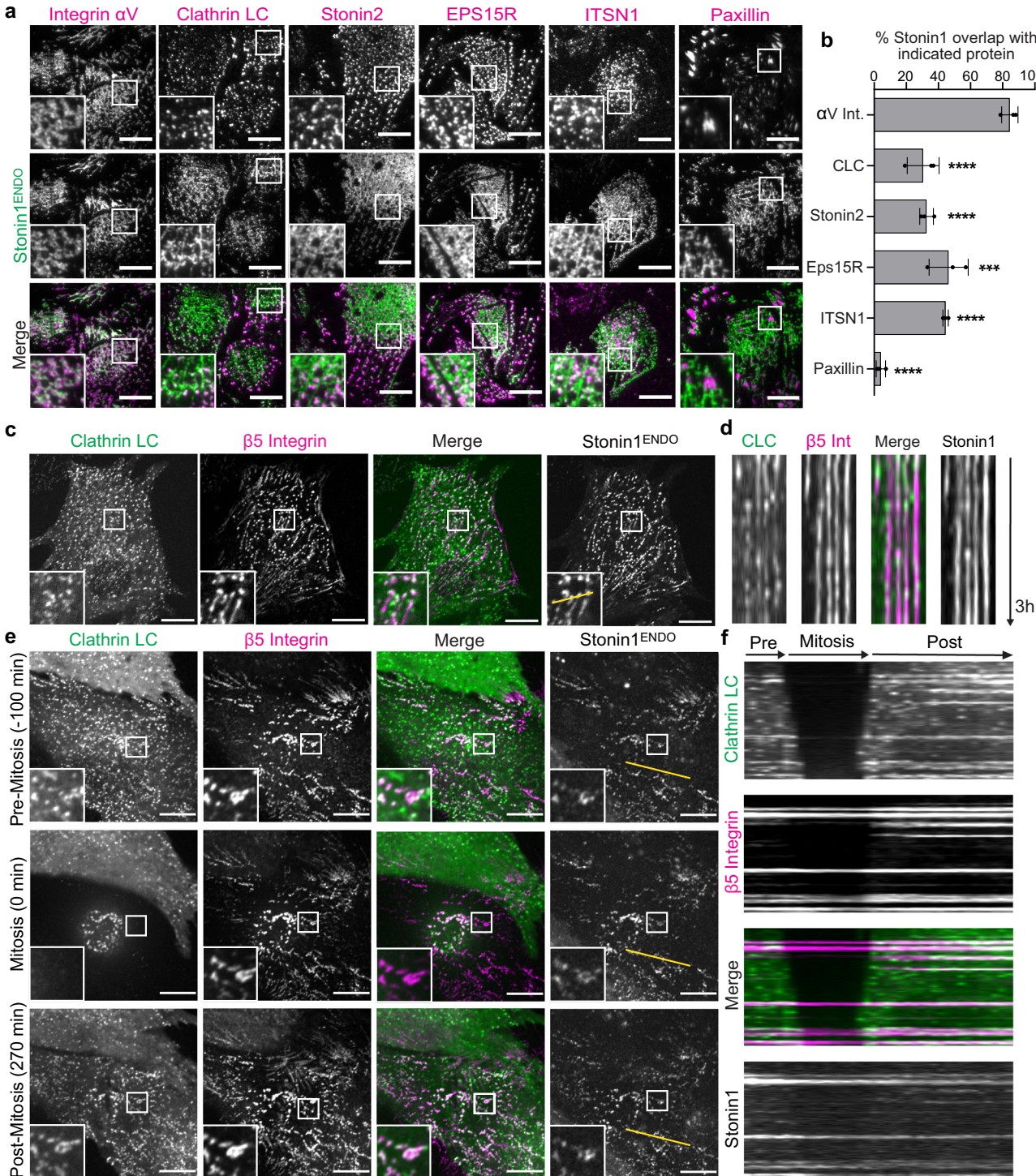

**Fig. 3 | αVβ5/stonin1-positive networks are not identical to RAs/plaques. a** αV integrin-positive adhesion networks show a large overlap with stonin1, but only a limited, mostly punctate co-localization with endocytic markers including clathrin. C2C12 cells expressing endogenously tagged EGFP-stonin1 were grown on collagen for 72 h, fixed and immunolabeled with antibodies against EGFP and the proteins indicated on top of the images. Scale bars, 10 μm. **b** Quantification of co-localization of stonin1 as αV integrin adhesion network marker with endocytic and focal adhesion proteins using the Mander's coefficient (mean±SEM, *N* = 3 independent experiments, One-way ANOVA with Dunnett´s multiple comparison test, ***$p$ = 0.0001, **** $p$< 0.0001). **c** In contrast to αVβ5 integrin and stonin1, clathrin is a much more short-lived component of αVβ5 adhesion networks. C2C12 cells endogenously expressing EGFP-stonin1 and stably transfected with mRFP-clathrin

light chain and β5 integrin-iRFP were seeded on collagen for 24 h and analyzed by live cell TIRF microscopy. Scale bars, 10 μm. **d** 3 h kymographs along the yellow line indicated in (**c**) reveal co-localizing long-lived stonin1 and β5 integrin structures in contrast to partially co-localizing much shorter-lived clathrin structures.
**e** Remodeling of external RAs/plaques during mitosis entails loss of clathrin. C2C12 cells expressing endogenously tagged EGFP-stonin1 were stably transfected with mRFP-clathrin light chain (LC) and β5 integrin-iRFP and subjected to TIRF live cell microscopy. Scale bars, 10 μm. See also Movie 1. **f** 370 min kymographs along the yellow line indicated in (**e**) reveal loss of clathrin from αVβ5 integrin scaffolds anchoring mitotic retraction fibers during mitotic cell rounding. Source data are provided as a Source Data file.

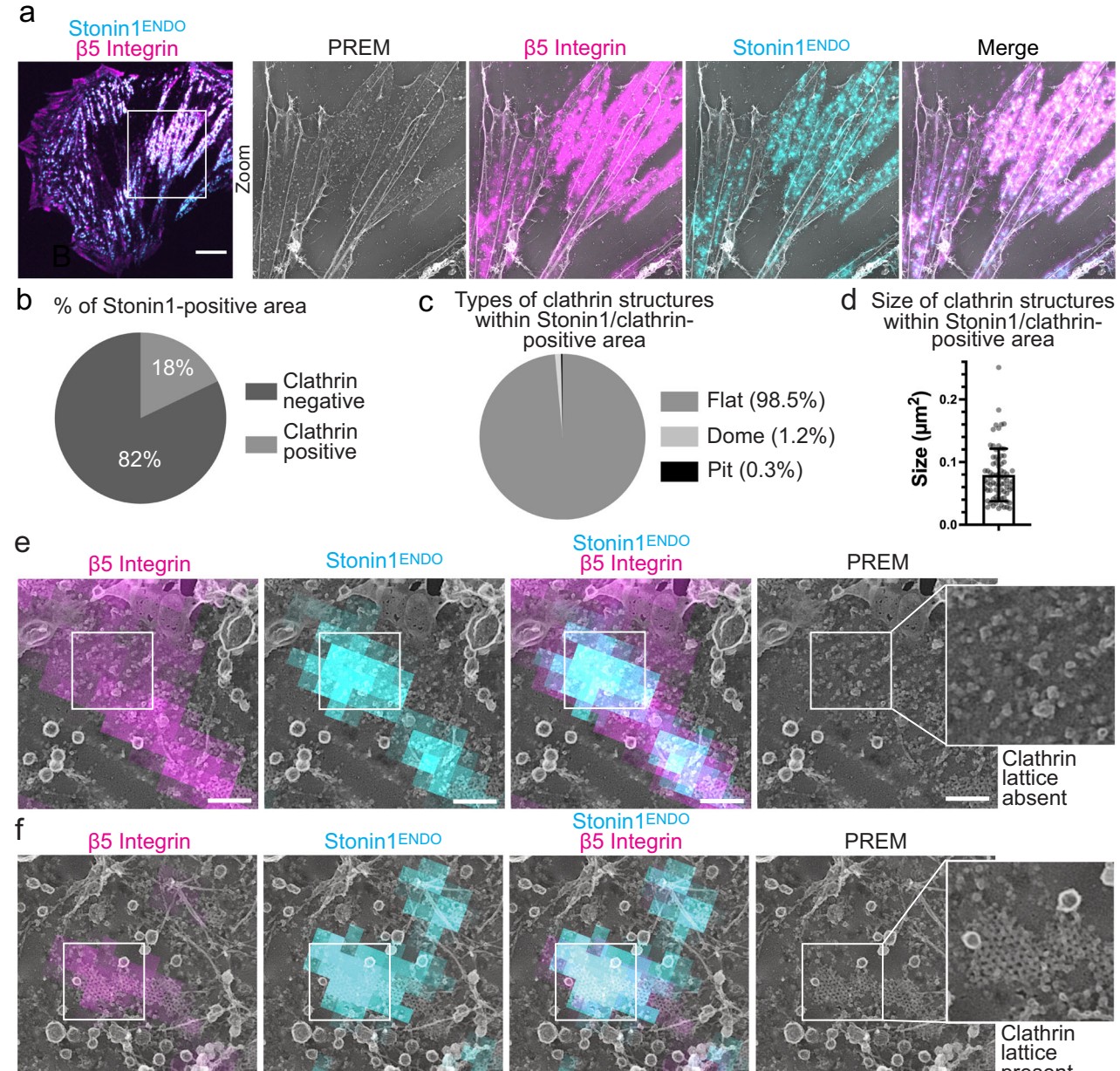

**Fig. 4 | αVβ5 integrin networks overlap only to a small extent with flat clathrin lattices.** C2C12 cells endogenously expressing EGFP-stonin1 and stably transduced with β5 integrin-iRFP were grown for 48 h on collagen and subjected to correlated light and electron microscopy after unroofing. **a** Example overview images. The white box in the fluorescence image represents the area observed by PREM. Scale bar, 10 μm. **b** Percentage of stonin1-positive adhesion network area that is positive for clathrin. The analysis comprised 617 stonin1-positive areas of which 133 were positive for clathrin. They were taken from 8 plasma membrane regions from four unroofed cells that were processed in 2 independent experiments. **c** Distribution of detected clathrin structures within stonin1-positive adhesion networks in the categories: flat clathrin lattices, dome-shaped clathrin cages and clathrin-coated endocytic pits. **d** Size distribution of clathrin structures detected within stonin1-positive adhesion networks (mean±SD, *n* = 76 clathrin structures from 2 independent experiments). **e** Example of frequent clathrin-free areas at sites of αVβ5/stonin1-positive adhesion network. Scale bar, 250 nm. **f** Example of rare flat clathrin lattice co-residing with αVβ5 integrin and stonin1. Scale bar, 300 nm. Source data are provided as a Source Data file.

The great challenge for the future will be to decipher the factors that control the transition between the different adhesion states by inducing the necessary alterations in the respective adhesion type-specific αVβ5 integrin interactome. While focal adhesion-induced modifications of the extracellular matrix have been proposed to be involved in the conversion of focal adhesions to RAs/plaques[18], it remains elusive how these alterations would result in differential integrin interactions. It is tempting to speculate that kinase-dependent posttranslational modifications might play a role since kinases have been shown to be involved in RA/plaque formation[17] and are known to influence αVβ5 integrin interactions and localization in different adhesion structures[13]. Intriguingly, stonin1 is also a heavily phosphorylated protein whose adhesion localization depends on its phosphorylation status. Clearly, additional studies will be needed to resolve the spatiotemporal regulation of integrin adhesion interconversion.

## Methods

### Plasmids and antibodies

Detailed information on plasmids is provided in Supplementary Data 1, and information on antibodies is summarized in Supplementary Data 2 and can also be found within the Reporting Summary file.

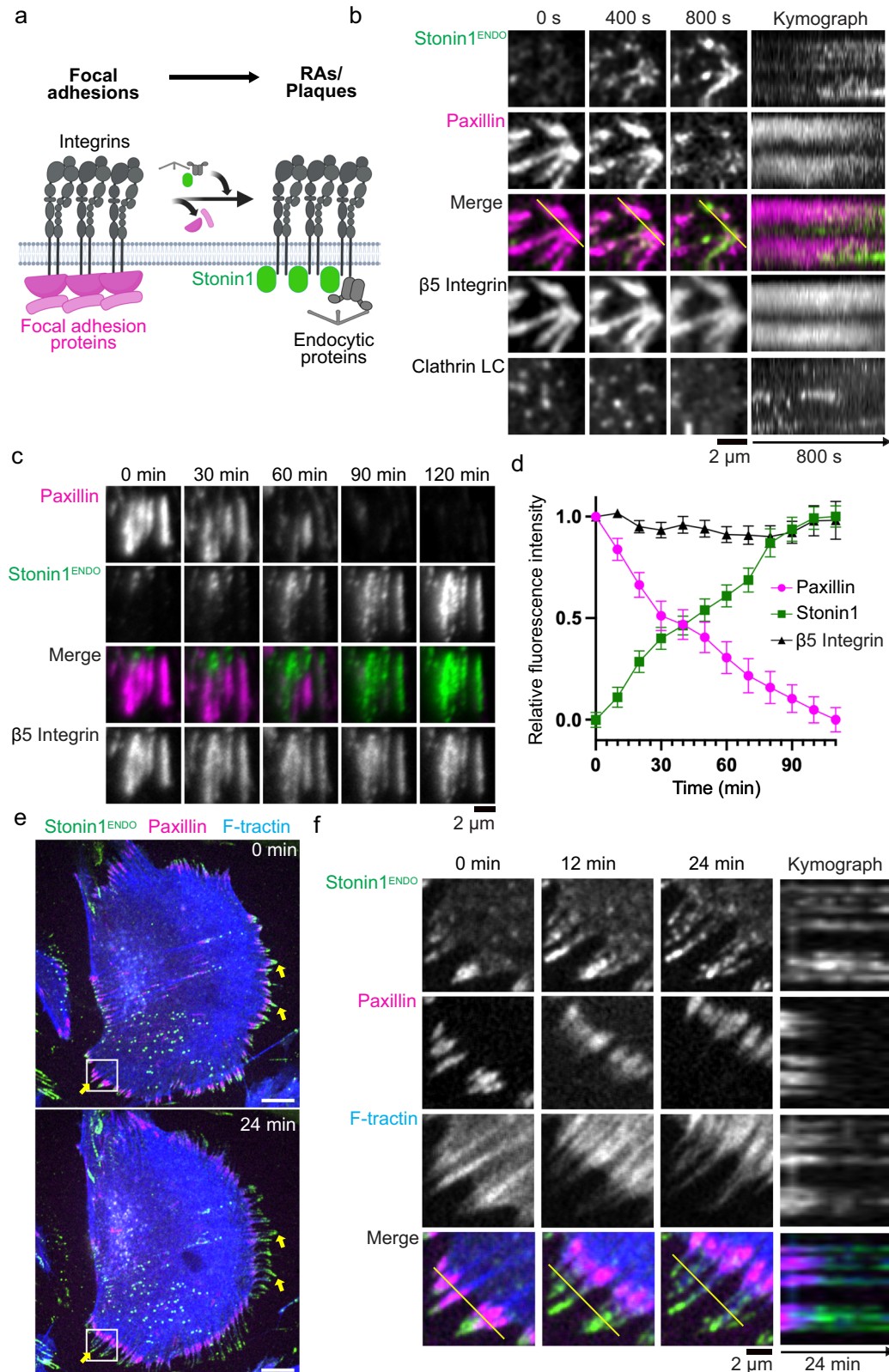

## Cell culture

Immortalized C2C12 mouse myoblasts (RRID: CVCL_0188), hTERT RPE1 (RRID: CVCL_4388) and HEK293T (RRID: CVCL_0063) cells were continuously cultured in DMEM containing 4.5 g/l glucose, 10% FCS, 100 U/l penicillin and 0.1 mg/ml streptomycin at 37 °C and 5% $CO_2$ for all experiments. Primary fibroblasts were prepared and cultured as described in ref. 23. Primary astrocytes from control and tamoxifen-inducible conditional AP2μ knockout mice[27] were prepared and cultured as described in ref. 25. Primary myoblasts were prepared and cultured as described in ref. 28,29.

## Coatings

For vitronectin coating, glass surfaces were incubated with 5 μg/ml vitronectin (Gibco, A14700) at 4 °C overnight. For Matrigel coating,

**Fig. 5 | Conversion of focal adhesions into non-canonical αVβ5 integrin adhesions. a** Scheme of how RAs/plaques can arise during focal adhesion disassembly by recycling of the existing αVβ5 scaffold. **b** C2C12 cells expressing genome-edited EGFP-stonin1 and stably transfected with β5 integrin-iRFP, eBFP2-paxillin and mRFP-clathrin light chain were cultured for 48 h and subjected to confocal live cell imaging. Still images showing a representative disassembling focal adhesion converting into an RA/plaque by losing paxillin and acquiring stonin1, and kymograph (along yellow line) of focal adhesion to RA/plaque conversion. Scale bar, 2 μm. **c, d** C2C12 cells expressing genome-edited EGFP-stonin1 and transiently transfected with β5 integrin-iRFP and mCherry-paxillin were cultured for 48 h and subjected to TIRF microscopy. **c** Still images showing a representative disassembling focal adhesion converting into an RA/plaque by losing paxillin and acquiring stonin1 (scale bar, 10 μm). See also Movie 2. Scale bar, 2 μm. **d** Quantification of relative

fluorescence profiles of indicated proteins within β5 integrin-positive adhesions over the course of 110 min (mean±SEM, *n* = 36 adhesions). The highest fluorescence value for each imaged protein was set to 1 and all other fluorescence values of the same protein expressed relative to it. **e–f** Migratory retraction fibers can be generated from focal adhesions via loss of focal adhesion proteins and parallel stonin1 recruitment. C2C12 cells expressing genome-edited EGFP-stonin1 and stably transfected with eBFP2-paxillin and mCherry-F-tractin were grown for 48 h and subjected to confocal live-cell imaging. **e** Overview image of cell extending retraction fibers at 0 min and 24 min. Scale bars, 10 μm. **f** Magnification of insets showing retraction of paxillin towards cell body and replacement by stonin1 resulting in a paxillin-negative retraction fiber, and kymograph (along yellow line) of focal adhesion-to-retraction fiber conversion. Scale bar, 2 μm. Source data are provided as a Source Data file.

---

they were incubated 2x with 5% Matrigel (Corning®, 356231) in Opti-MEM for 10 min. Fibrillar collagen matrices were prepared similarly to ref. 30. 3.66 μg/ml rat tail collagen I (Corning, 354236) was first mixed 1:10 with ice-cold 10X DMEM and then 1:10 with 10X reconstitution buffer (26 mM NaHCO$_3$, 20 mM HEPES). The collagen solution was centrifuged for 3 min at 9000 g and 4 °C and dispensed with a pipette tip onto pre-cooled coverslips. The coverslips were incubated on ice for 5 min followed by 30 min at 37 °C and 5% CO$_2$.

### Western Blot
A total of one million cells were seeded onto 6 cm dishes coated with Matrigel or collagen I, and cultured overnight in Dulbecco's Modified Eagle Medium (DMEM) supplemented with 10% fetal calf serum (FCS). Cells were washed in cold PBS and lysed in lysis buffer [20 mM HEPES (pH 7.5), 150 mM NaCl, 2 mM EDTA (pH 7.5), 1% NP-40, 0.1% SDS] supplemented with 15 mM NaF, protease inhibitor cocktail (Sigma-Aldrich) and phosphatase inhibitor 2 & 3 cocktail (Sigma-Aldrich). Lysates were incubated on ice for 10 min and cleared by centrifugation at 14,000 g for 10 min at 4 °C. Samples were denatured in LDS Sample Buffer (Invitrogen) containing 50 mM DTT for 10 min at 80 °C. Proteins were separated by SDS-PAGE using Bolt Novex 4–12% gradient Bis-Tris gels (Invitrogen), transferred to nitrocellulose membrane (Amersham) and blocked for 60 min in 5% BSA in TBST buffer [10 mM Tris (pH 7.5), 150 mM NaCl, 0.1% Tween 20]. Incubation with primary antibodies diluted in 5% BSA in TBST buffer took place for 120 min at room temperature. After washing 3x 5 min with TBST membranes were incubated for 1 h at room temperature with IRDye-conjugated secondary antibodies and washed 3x 5 min with TBST. Signal intensities were quantified using Licor Image Studio.

### Blebbistatin treatment and wash-out
(-)-Blebbistatin (Selleckchem, S7099) was dissolved in DMSO to 10 mM and applied to cells in DMEM at 50 μM for 1 h. For wash-out, the DMEM was replaced with fresh serum-containing culture medium for 15 min.

### Transfection and RNA interference
C2C12 cells were transiently transfected at 30-40% confluency with JetPrime (VWR) according to the manufacturer's instructions. After 4 h incubation, the medium was changed. For RNAi C2C12 cells were reverse-transfected with INTERFERin (VWR) at low confluency according to the manufacturer's instructions. 20,000 cells per 12well were reverse-transfected with 25 nM siRNA. MISSION® siRNA (Sigma, SIC001) was used as a negative control and a 25 nM SMARTpool for silencing β5 integrin (Dharmacon, L-042453). Experiments were performed after 96 h.

### Virus production and transduction
For viral production, 10 cm dishes of HEK293T cells with a confluency of 60–80% were transfected with 15 μg transfer plasmid, 4.5 μg envelope plasmid pMD2.G and 10.5 μg packaging plasmid psPAX2

(lentivirus) or pCIG3.NG (retrovirus) using calcium phosphate-based transfection. Virus-containing medium was replaced with new medium at 16, 48 and 72 h and collected after 48 and 72 h. Cell debris was removed by centrifugation at 200× *g* for 5 min at room temperature, and the medium was stored at 4 °C until the final harvest. All harvested media were pooled, filtered with a 0.45-μm Falcon filter and concentrated 100x using an Amicon-15 100-kDa filter column (Merck, Z740210). Aliquots containing 20% of the concentrated viral supernatant were stored at -80 °C. For transduction, 100,000 cells seeded the previous day in 6well plates were transduced with the contents of one aliquot. Since the transfer plasmids encode a puromycin resistance gene, cells were selected after 96 h with 1 μg/ml puromycin to obtain stable cell lines. Point mutants of integrin β5 S759/S762 were generated by in vivo assembly[31] with the PCR-based overlap of primers containing the point mutants using Phusion DNA polymerase (Invitrogen) and β5 Integrin-iRFP source vector.

### Genome editing
To N-terminally tag the endogenous stonin1 with EGFP, a donor vector for homology-directed repair encoding EGFP flanked by sequences homologous to the STON1 locus was generated. In addition, a suitable guide RNA (designed by the CRISPR gRNA Design tool, Atum, Newark) resulting in a double-strand break close to the STON1 start codon was cloned into the BbsI site of the px458 plasmid (encoding Cas9-EGFP) using the forward primer CACCGATTCTACAAACCCGGGCAGC and the reverse primer AAACGCTGCCCGGGTTTGTAGAAT. The donor vector and the px458 vector expressing Cas9-EGFP and STON1-specific sgRNA were co-transfected with a 5x molar excess of the donor vector into C2C12 myoblasts with JetPrime (VWR). After 48 h the EGFP expressing cells were FACS-sorted and transferred as single cells into 150 μl conditioned medium within five 96well plates. When cells had proliferated suffiently, they were screened by confocal microscopy for the presence of EGFP-stonin1 expression and verified by Western blot resulting in the identification of a heterozygous clone which was used for this study.

This clone was used as a parental cell line for the endogenous tagging of integrin β5. To C-terminally tag the endogenous integrin β5 with mScarlet3, a donor vector for homology-directed repair encoding mScarlet3 flanked by sequences homologous to the integrin β5 locus was generated. In addition, a suitable guide RNA resulting in a double-strand break close to the integrin β5 stop codon was cloned as above into the px458 plasmid using the forward primer CACCG-TAGTCCCCCTCCAGCCATCC and the reverse primer AAACG-GATGGCTGGAGGGGGACTA. The donor vector and the px458 vector expressing Cas9-EGFP and integrin β5-specific sgRNA were co-transfected as above into C2C12 cells, subsequently FACS-sorted and transferred as single cells into 150 μl conditioned medium into 96-well plates. After sufficient proliferation cells were screened as above for the presence of integrin β5-mScarlet3. By PCR we identified a heterozygous clone which was used for this study.

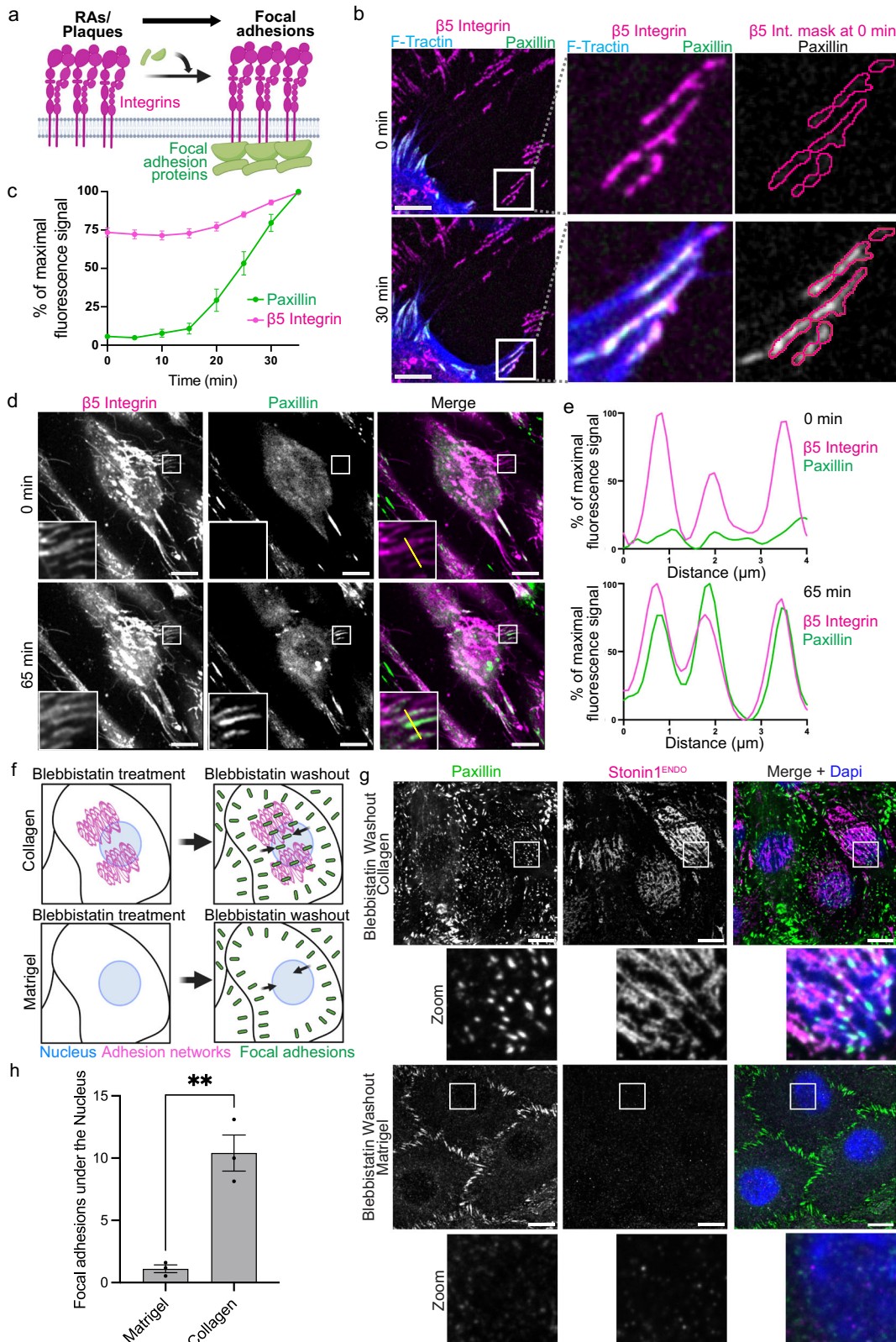

## Immunofluorescence

Cells grown on coverslips or cell culture slides (Greiner, 543079) were rinsed 1x with PBS and fixed with ice-cold 4% paraformaldehyde (PFA) containing 4% sucrose in PBS for 10 min at room temperature. After washing 3x with PBS, cells were simultaneously permeabilized and blocked in goat serum dilution buffer (GSDB; 0.1% TritonX-100, 10% goat serum in PBS) for 30 min. Antibodies were diluted in GSDB (for

dilutions see Supplementary Data 2) and applied for 1 h. After washing 3x with PBS, cells were incubated with secondary antibodies (for dilutions see Supplementary Data 2) in GSDB for 45 min. When labeling F-actin, 8.25 μM phalloidin conjugated with AlexaFluor488 (Invitrogen/A12379) or Alexa Fluor568 (Invitrogen/A12380) dissolved in methanol was added during the secondary antibody application at 1:50. After rinsing 3x with PBS, cells were incubated with 1 μg/ml DAPI (Sigma/

**Fig. 6 | Conversion of non-canonical αVβ5 integrin adhesions into focal adhesions. a** Scheme of how focal adhesions can arise from non-canonical αVβ5 integrin adhesions by recruiting focal adhesion proteins. **b–c** Focal adhesions assemble at β5 integrin-positive migratory retraction fibers. C2C12 cells stably expressing EBFP2-paxillin, mCherry-F-tractin and β5 integrin-iRFP were grown for 48 h and subjected to confocal live-cell imaging. **b** Representative images, scale bars, 10 μm. Magnification of grey inset at 0 min shows paxillin-negative β5 integrin-positive retraction fibers left behind by migrating cell. Magnification of inset at 30 min shows recruitment of focal adhesion marker paxillin to β5 integrin scaffold upon cellular respreading across retraction fiber. See also Movie 3. **c** Quantification of normalized fluorescence profile of EBFP2-paxillin and integrin β5-iRFP within the β5 integrin mask shown in (**b**) over time (mean±SEM, 15 adhesions from N=3 independent experiments). The highest fluorescence value for each imaged protein over the time course was set to 100%, and all other fluorescence values of the same protein were expressed relative to it. **d,e** Focal adhesions assemble at β5 integrin-positive mitotic RAs/plaques. C2C12 cells stably expressing EBFP2-paxillin and β5 integrin-iRFP were grown for 48 h and subjected to confocal live cell imaging. **d** Magnified insets at 0 min show β5 integrin-positive mitotic RAs/plaques of a cell in cytokinesis which has disassembled its focal adhesions. Magnified insets at

65 min show recruitment of focal adhesion marker paxillin to former mitotic RAs/plaques converting them into focal adhesions. Scale bars, 10 μm. See also Movie 4. **e** Normalized intensity profile (along the yellow line depicted in (**d**)) of EBFP2-paxillin and β5 integrin-iRFP at cytokinesis (0 min) and during respreading (65 min). The highest fluorescence value for each imaged protein along the line and across both time points was set to 100%, and all other fluorescence values of the same protein were expressed relative to it. Scale bars, 10 μm. **f–h** Focal adhesions can assemble internally at αVβ5 integrin adhesion networks. **f** Scheme illustrating experiment. **g** C2C12 cells endogenously expressing EGFP-stonin1 were seeded on collagen to promote αVβ5 integrin network formation or on Matrigel to suppress it. Cells were treated for 1 h with blebbistatin to disassemble focal adhesions. After 30 min of blebbistatin washout, cells were fixed, immunolabeled for EGFP and paxillin and analyzed by TIRF microscopy. Nuclei were stained with DAPI. Scale bars, 10 μm. Focal adhesions can be seen to form internally at sites of αVβ5 integrin adhesion networks. In absence of αVβ5 integrin adhesion networks, focal adhesion formation is observed in the cell periphery, but hardly centrally. **h** Quantifcation of segmented paxillin puncta with a pixel intensity >10 in the area outlined by DAPI (mean±SEM, $N = 3$ independent experiments, two-tailed unpaired Student´s t-test, **$p < 0.01$). Source data are provided as a Source Data file.

D9542) for 5 min and then rinsed 2x with PBS. Glass coverslips were mounted on slides using ImmuMount™ (Thermo Fisher, 9990402). For TIRF microscopy, cell culture slides were not mounted but kept in 1x PBS and imaged on the same day.

## Microscopy

Confocal imaging was performed with a Nikon spinning disk confocal microscope equipped with an Okolab environment control chamber for life cell imaging at 37 °C and 5% $CO_2$ and a Nikon PerfectFocus autofocus system. A 60x (PLAN APO, NA: 1.40, WD 0.13 mm) oil immersion objective was used. The setup was controlled by the imaging-Software NIS (Nikon). Live cell imaging was performed in imaging buffer (FluoroBrite DMEM with 4.5 g/l glucose, 10% FCS, 1x GlutaMAX™ supplement, 100 U/l penicillin, 0.1 mg/ml streptomycin). TIRF microscopy was performed using the Nikon equipment described above with an 100x immersion objective (Apochromat TIRF, NA 1.49, WD 0.12 mm). FRAP was also performed on the Nikon spinning disk confocal microscope. Briefly, a TIRF image was acquired before bleaching, enabling robust recovery standardization. Both, non-canonical adhesions and focal adhesions were then bleached in confocal mode using 5% of maximal 488 nm laser power for 1 s. Recovery was then monitored in TIRF mode for intervals of 4 min. Acquisition and switching between confocal and TIRF mode (>2 s) was automated using NIS-Elements software. Three sequential images were taken per experiment. Microscopy images were imported and analyzed using ImageJ/Fiji (Bioformats importer).

## Correlative fluorescence and platinum replica transmission electron microscopy (PREM)

For correlative fluorescence and electron microscopy, C2C12 myoblasts endogenously expressing EGFP-stonin1 and stably transduced with β5 integrin-iRFP were seeded on gridded glass coverslips (Ibidi, #10816) that were coated with collagen. Before unroofing, cells were washed 2x with stabilization buffer (5 mM $MgCl_2$, 70 mM KCl, 30 mM HEPES, pH7.4). 16 h postseeding unroofing was performed in 2% paraformaldehyde (PFA) in stabilization buffer using a 10-ml 22-gauge, 1.5-inch syringe (BD Biosciences #309604) to splash the cover slips positioning the needle -1 cm from the coverslips. Afterwards, the resulting membrane sheets were immediately transferred into fresh 4% PFA and fixed for 20 min at room temperature. After fixation, the membrane sheets were washed 3x with PBS and incubated with AF594-conjugated phalloidin (Biomol, ABD-23158) for 30 min at 1:1000. After another three PBS washes the membrane sheets were immediately used for spinning disk confocal microscopy. To cover a defined grid of the coverslip, a montage of 12x12 images was taken with a 60x

immersion objective (PLAN APO, NA 1.40, WD 0.13, 110 nm pixel size). Afterwards, unroofed membranes were fixed overnight with 2% glutaraldehyde in PBS. Subsequently, the samples were stained with 1% tannic acid (EM grade), 0.1% uranyl acetate and dehydrated with 15–100% ethanol. Afterwards, the membrane sheets were dried using $CO_2$ critical point drying (Leica) followed by 3 nm platinum and 5.5 nm carbon coating (Leica ACE600) as described in ref. [32]. The coverslip was removed by 4% hydrofluoric acid. Membrane replicas were washed extensively with water and then transferred to 70mesh copper TEM grids (Science Services) coated with Formvar for imaging with TEM Zeiss 900 at a magnification of 20,000 times (pixel size 1.5 nm). Montage PREM images were acquired manually and stitched together in Adobe Photoshop. MATLAB was used to correlate fluorescence images with montage PREM images as described in[32,33].

## Image analysis

Image analysis was performed with the open-source software Fiji[34]. Cells for analysis were randomly selected according to the fluorescence signal from the nuclear stain. To quantify co-localization between two channels, Pearson correlation coefficients were determined using the Fiji plugin 'Coloc2' with ' Threshold regression'. Prior to image processing, the background was manually removed by measuring pixel intensities in a black area, and background noise was removed by applying a Gaussian filter. The same was done for determining the Mander's overlap coefficient. Here the Fiji plugin "JACoP"[35] was used, and the threshold was set manually. Image analysis of large image sets was performed automatically with the modular high-throughput image analysis software CellProfiler[36]. For segmentations of particles or cells, the lower quartile of intensities was removed automatically as background, and automatic thresholding was performed using the Otsu algorithm. Mean fluorescence intensities were measured using either automatically detected particle or cell segmentations or by manually selecting a region of interest and measuring it using the measurement tool in Fiji. Recursive alignment (registration) of multichannel hyperstacks in life cell imaging movies was performed with the ImageJ/Fiji plugin HyperStackReg that is based on StackReg[37].

## Statistics and Reproducibility

Data are depicted as means±SEM if not otherwise indicated. For comparisons between two experimental groups, statistical significance was evaluated using two-tailed unpaired Student's t-test if data were normally distributed. Where more than two groups were compared, One-way ANOVA with either Dunnett´s or Tukey´s multiple comparison test (as indicated in the legend) was used for normally distributed

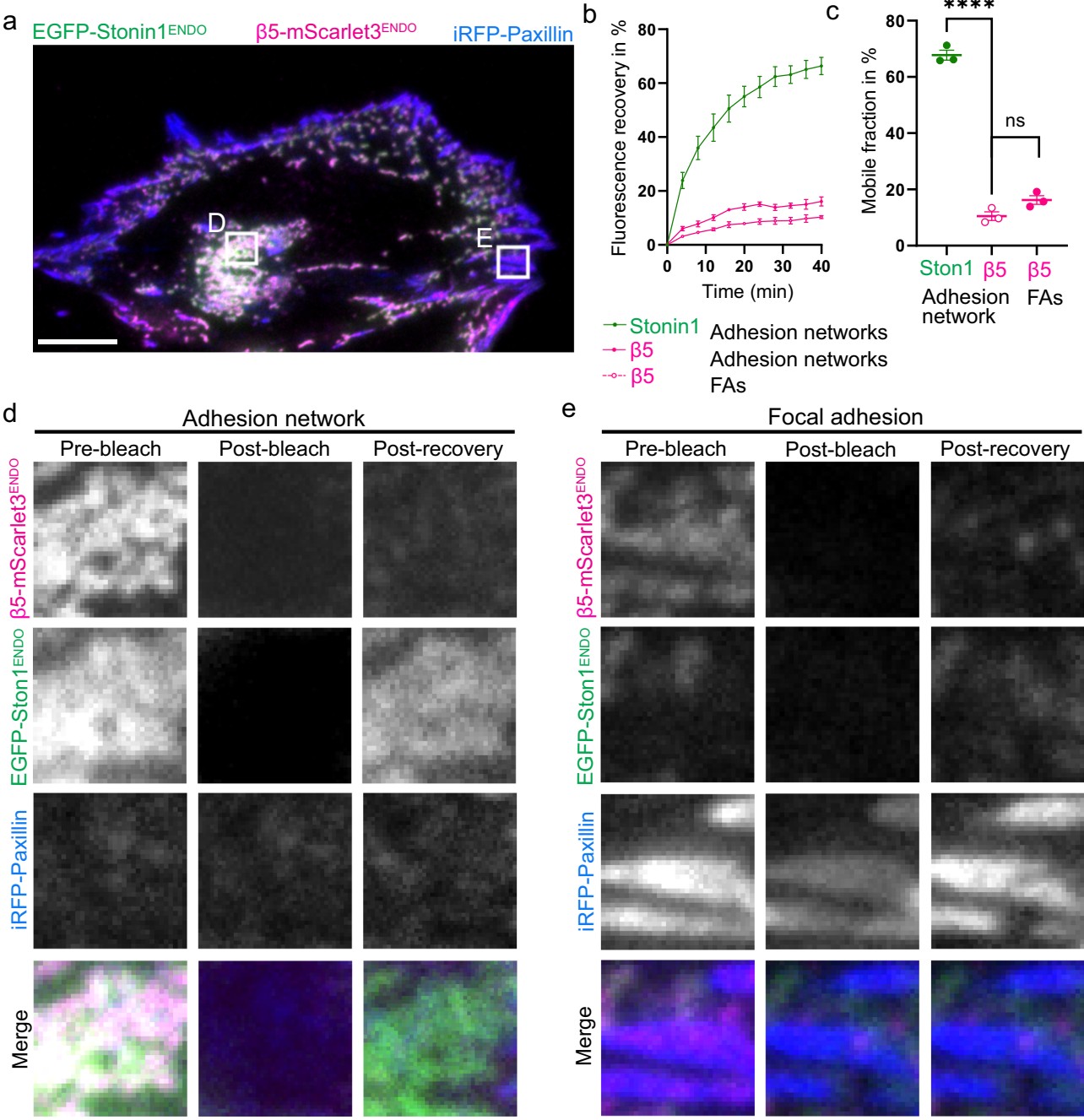

**Fig. 7 | β5 integrin molecules display low turnover within adhesions in comparison to the quickly exchanging stonin1 proteins.** (**a**) C2C12 cells expressing genome-edited EGFP-stonin1 and β5 integrin-mScarlet3 were transduced with iRFP-paxillin and cultured for 48 h. Scale bar, 10 μm. Parts of large stonin1- and β5 integrin-positive adhesion networks (example in **d**) and areas comprising paxillin- and β5 integrin-positive focal adhesions (example in **e**) were photobleached, and the fluorescence recovery after photobleaching (FRAP) of EGFP-stonin1 and β5 integrin-mScarlet3 was monitored for 40 min (**b**). **c** Quantification of the recovered,

that is, mobile fraction of EGFP-stonin1 and β5 integrin-mScarlet3 molecules based on an exponential fit of the FRAP data reveals a low mobile pool for the β5 integrin scaffold in comparison to the accessory protein stonin1 **b, c** mean±SEM, $N = 3$ independent experiments with each comprising three movies capturing two cells where one canonical and one non-canonical adhesion per cell was photobleached; **c** One-Way ANOVA with Tukey´s multiple comparison test, ****$p < 0.0001$; ns, nonsignificant). Source data are provided as a Source Data file.

data. Where data had to be normalized before analysis, one-sample t-tests were used for comparisons with control group values. For comparing more than two groups, the Kruskall-Wallis test with Dunn´s multiple comparison test was used. Significance levels are indicated as *$p < 0.05$, **$p < 0.01$, ***$p < 0.005$, and ****$p < 0.0001$. Differences that are not significant are indicated as ns. Statistical data evaluation was performed using GraphPad Prism 9.3.1. Results depicted in the form of representative images, including images of adhesion types and

microscopical pictures of life-cell imaging experiments, were observed in at least three independent experiments and were also verified with a cell line gene-edited to express mScarlet3-tagged integrin β5 from its endogenous locus.

### Reporting summary

Further information on research design is available in the Nature Portfolio Reporting Summary linked to this article.

## Data availability
All data underlying the findings presented in this paper are either contained within the paper or its supplementary files. In addition Source data are provided with this paper.

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

## Acknowledgements
We thank Claudia Schmidt for technical assistance. Schemes were created with BioRender.com. This research was funded by grants of the Deutsche Forschungsgemeinschaft (DFG, German Research Foundation) to T.M. (project numbers MA 4735/2-1 and 461336323). C.M. was supported by the Intramural Research Program of the National Heart Lung and Blood Institute, National Institutes of Health, USA (Lab of Justin W. Taraska).

## Author contributions
FL and TM conceived the project and designed the experiments. FL performed microscopy and biochemical experiments. NS performed experiments with β5 integrin mutants. IL provided primary myoblasts. TLH and VH provided primary astrocytes. ML and JP assisted with high resolution microscopy. CM and DP performed and assisted with correlative metal replica electron microscopy. TM prepared the manuscript with input and approval from all authors.

## Funding

## Competing interests

The authors declare no competing interests.
