## [Peer Review File · Nature Communications]

Canonical and non-canonical integrin-based adhesions dynamically interconvertReviewer #1 (Remarks to the Author):

Overview:

This manuscript investigates and compares the molecular composition, morphology and interconvertibility of canonical (e.g. focal adhesions, FAs) and non-canonical (e.g. reticular adhesions, RAs, clathrin plaques etc) cell-matrix adhesions mediated by the integrin $\alpha_5\beta_1$. The major advances delivered herein are: a) demonstrating the utility of stonin1 as a selective marker for non-canonical adhesions (and not for endocytic structures as previously thought) – a key prerequisite to further developments in this research field, and: b) showing that canonical and non-canonical adhesions can dynamically and bidirectionally interconvert (as well as forming distinctly, *de novo*). These represent important incremental advances in understanding of recently discovered non-canonical adhesion classes. These advances will facilitate and direct subsequent research interrogating these structures as dynamic constituents of the $\alpha_5\beta_1$ -mediated adhesion 'ecosystem'. Given these advances, and a rapidly growing audience for this research, I support the suitability of this article for Nature Communications.

Detailed Comments:

1) When cells are cultured on specific ECM substrates, e.g. vitronectin in Figure 1 or collagen in Figure 4, are the cells grown with or without FCS? This should be indicated in legends and/or methods, to give the reader a sense of the likely purity of the ECM condition. Growth on specific substrates in the presence of FCS is, to my understanding, unlikely to produce results reflective of pure ECM substrates (since FCS is full of vitronectin, fibronectin etc etc), unless some mechanism of surface occlusion has been performed – which I cannot detect in the manuscript.

2) Figure 5 explores the dynamics of interconversion from FAs to RAs. This is an important finding (which incidentally comports with observations of bi-directional inter-convertibility not published with the original RA NCB paper). It is noted herein that β_5 levels remain largely unchanged during the transition from paxillin-positive to stonin1-positive. It would be interesting to assess β_5 integrin turnover kinetics around this transition (or, perhaps more feasibly, in adjacent adhesion structures that are paxillin- versus stonin1-positive). Given the reported differences in turnover rate / extent, it would be interesting to see if this difference aligns closely to the molecular markers used herein (paxillin vs stonin). Differences in integrin dynamics may be a fundamental 'biomarker' for differences in adhesion complex structure/function.

3) There is some occluded text on the right side of Fig 5 Panel D.

4) The observation of paxillin recruitment to existing non-canonical adhesions (e.g. retraction fibres), described in Figure 6, is interesting, and also comports with observations made (but not published) with the original RA NCB paper (and similarly, the formation (nucleation?) of small canonical adhesion punctae on the edge of / immediately adjacent to existing reticular adhesions, as shown clearly in Fig 6G). However, can the authors clarify in text when / if they are referring to a build up of e.g. paxillin, on non-canonical adhesions derived from a different cell (at any point in this section). While rapid build up of canonical components on non-canonical (e.g. retraction fibre) adhesions from the same cell is easy to understand, how 'orphan' integrins (and membrane?) attached to a surface (and external to the cell in question) can be used as a scaffold by cytoplasmic proteins, would seem to require some explanation. Do surface-residual membranes fuse with a passing cell, making the integrins in the residual membranes part of / accessible to the passing cell (and cytoplasmic proteins within that passing cell)? How might this otherwise be possible? This should at least be discussed, since it seems to imply some additional biophysical processes that are not yet considered herein. Unless I have missed something? Perhaps the emerging literature on migrasomes left as 'cues' by cells in 2D / 3D culture may provide some insights?

Reviewer #2 (Remarks to the Author):

In this manuscript entitled "Canonical and non-canonical integrin-based adhesions dynamically interconvert", Lukas et al. identify a protein marker called stonin1, which plays a role in the interconversion of canonical and non-canonical adhesions in conjugation with $\alpha V\beta 5$ integrin. The authors find that the stonin1- $\alpha V\beta 5$ integrin complex acts as an intermediate and decides the fate of the formation of canonical adhesions and non-canonical adhesions. The authors propose that non-canonical adhesive structures act as points of origin for generating canonical focal adhesions.

Although this work focuses on an important topic and overall, the experimental design/analysis is of good quality, my enthusiasm is diminished by the fact that the message describing the conversion of plaques into focal adhesions has been already published in Zuidema et al., 2022 J Cell Sci while two additional preprints show it as well (Bucher et al., bioarxiv doi: <https://doi.org/10.1101/493114>; and in Hakanpää et al., bioarxiv doi: <https://doi.org/10.1101/2022.05.30.494022>). Some conclusions drawn such as the exclusion of stonin1 and $\beta 5$ integrin from plaques are sometimes too generalized, lack molecular biochemical mechanisms, and are not controlled enough. For example, the present study lacks mechanistic insight into how modifications of $\alpha V\beta 5$ integrin affect the affinity and binding interactions with stonin1 and favor the formation of different adhesive structures. Also, the entire paper focuses on undifferentiated C2C12 myoblasts (this cell line is known to be a poor model for clathrin plaque formation) while plaques have been extensively characterized in differentiated myotubes as their number sharply increases after differentiation.

Major comments:

1/ One of the major claims concerns the successive transition from focal adhesions to RAs/plaques without altering the integrin scaffold. Although this observation is interesting, $\alpha V\beta 5$ localization pattern is regulated by the amount of cellular tension, it has already been shown by the Sonnenberg group (Zuidema et al., 2018) that when tension is increased, integrin $\alpha V\beta 5$ is localized predominantly in FAs and switch to FCLs when cellular tension is reduced. Here the authors have not taken into consideration how cellular tension affects the stonin1 localization in conjugation with $\alpha V\beta 5$, while this might seem trivial, in the particular work it is a major issue, as it defines the interplay between canonical and non-canonical adhesions.

2/The PREM correlative imaging in Fig. 4 is a strong argument in favour of the fact that stonin1 and integrin beta5 are not exclusively localized in clathrin lattices. However, the authors should perform additional controls. For instance, $\beta 5$ is always overexpressed and could be deleterious to the localization of clathrin or other components. The authors should at least analyse the distribution of endogenous $\beta 5$ integrin using antibodies instead of overexpressing it stably.

3/The authors show regions where Stonin1 and $\beta 5$ integrin colocalize but no clathrin is present. However, the EM images are not clearly showing what is below the fluorescent signal. Are these clumps of overexpressed $\beta 5$? Are there any vesicles? This is an important point considering the fact that previous work has shown localization of $\beta 5$ in clathrin lattices and Ras. Have the CLEM data been validated with immunogold labeling the endogenous proteins?

4/ On line 202 the authors state "First, our experiments in migrating cells confirmed that RAs/plaques can form denovo, although such events appeared to be rare (Fig. S5A-C)". Here in this context, the authors need to provide kymographs of clathrin assembly in comparison with stonin1. Here the authors have not taken into consideration endogenous $\alpha V\beta 5$ integrin contributions in these RAs/Plaques. Quantification of these events would be important before generalizing the phenomena.

5/ The present study focuses on endogenously tagged EGFP-stonin1, but stably overexpressed $\beta 5$ -integrin throughout the study. Expression levels of these proteins play a role in the stonin1- $\alpha V\beta 5$ integrin scaffold. The authors need to provide endogenously tagged $\beta 5$ -integrin that doesn't affect the expression levels of all the mentioned canonical and non-canonical markers. The authors should control this by showing that the tagged and untagged stonin 1 level doesn't affect clathrin, and actin expression levels by using western blot.

6/ The study focuses exclusively on C2C12 cells, but what about other cell types, formation of

canonical and non-canonical adhesions differ in various cell types. It is very important to test stonin1 behavior in other cell types before considering the novel marker for both canonical and Reticular adhesions/clathrin plaques. Have the authors tried using Hela cells which also form plaques?

7/ The present study focuses on RAs/plaques that can also arise during focal adhesion disassembly by recycling the existing $\alpha\text{V}\beta 5$ scaffold. It is very important to test this for instance by using FRAP experiments to show the same $\alpha\text{V}\beta 5$ facilitates interacting with stonin1 and canonical protein markers before generalizing the proposed model scenario.

8/ On line 253 the authors state that "However, in presence of non-canonical adhesion networks (i.e. on collagen), small focal adhesions appeared also centrally right next to the networks (Fig. 6G, H) suggesting again a novel mode of focal adhesion assembly that rather depends on pre-existing $\alpha\text{V}\beta 5$ integrin scaffolds than on lamellipodial protrusion". The authors need to provide CLEM ultra-structural views on such conditions i.e. synchronized focal adhesion assembly events adjacent to non-canonical adhesive structures.

9/ The present study focuses on endogenously tagged EGFP-stonin1, but overexpression of beta5 integrin which could perturb the clathrin homeostasis and therefore contribute to the difference between this work and previously published papers on plaques showing the presence of endogenous $\beta 5$ integrin in clathrin plaques.

10/ On line 636 the authors state "Percentage of stonin1-positive adhesion network area that is positive for clathrin (N=4 unroofed cells)". The number of cells but not the number of independent replicates and the number of clathrin structures considered for the analysis. One should know statistics on these structures, and also how big are these stonin1 and clathrin-positive structures. These statistics give a morphometric analysis of these ultra-structures before coming to conclusion. Have these experiments in Figure 4 and Fig S4 been performed only once?

11/ Clathrin lattices were detected throughout the manuscript with antibodies against the clathrin light chain. Does this antibody recognize both light chains isoforms CLCa and CLCb? The authors should at least once validate the absence of clathrin in beta5/stonin1 regions with a clathrin heavy chain antibody which might also prove more reactive.

Minor comments:

-Typo in the sentence in line 58 "heterdimers, provide the molecular link between RAs/plaques and the extracellular matrix" needs to be changed to heterodimers.

The legends in several panels (i.e. Figure 5D and Figure 6E) are not clear enough.

Manuscript NCOMMS-23-09818 - Point-by-point response to the reviewer's comments

Reviewer #1

This manuscript investigates and compares the molecular composition, morphology and interconvertibility of canonical (e.g. focal adhesions, FAs) and non-canonical (e.g. reticular adhesions, RAs, clathrin plaques etc) cell-matrix adhesions mediated by the integrin $\alpha 5 \beta 1$. The major advances delivered herein are: a) demonstrating the utility of stonin1 as a selective marker for non-canonical adhesions (and not for endocytic structures as previously thought) – a key prerequisite to further developments in this research field, and: b) showing that canonical and non-canonical adhesions can dynamically and bidirectionally interconvert (as well as forming distinctly, *de novo*). These represent important incremental advances in understanding of recently discovered non-canonical adhesion classes. These advances will facilitate and direct subsequent research interrogating these structures as dynamic constituents of the $\alpha 5 \beta 1$ -mediated adhesion 'ecosystem'. Given these advances, and a rapidly growing audience for this research, I support the suitability of this article for Nature Communications.

We thank the reviewer for his/her appreciation of the important advances that our study contributes to our understanding of non-canonical adhesions and for considering our manuscript suitable for publication in Nature Communications.

In our revised manuscript we have addressed all remaining points the reviewer raised. For easier tracking of the changes we coloured the new text in the pdf file of the revised manuscript and supplement in red.

Detailed Comments:

1) When cells are cultured on specific ECM substrates, e.g. vitronectin in Figure 1 or collagen in Figure 4, are the cells grown with or without FCS? This should be indicated in legends and/or methods, to give the reader a sense of the likely purity of the ECM condition. Growth on specific substrates in the presence of FCS is, to my understanding, unlikely to produce results reflective of pure ECM substrates (since FCS is full of vitronectin, fibronectin etc etc), unless some mechanism of surface occlusion has been performed – which I cannot detect in the manuscript.

For all experiments cells were continuously cultured in the presence of 10% FCS as we now stress in the *Cell Culture* part of the methods section. We tested also FCS-free cultures, but did not observe any large non-canonical adhesion structures in absence of serum, suggesting that certain components in FCS are essential for their biogenesis. Among these components are likely growth factors in line with data from the Taraska lab which shows that clathrin structures start to grow into large flat plaques upon growth factor addition (Alfonzo-Méndez et al, 2022; doi.org/10.1038/s41467-022-28373-x). Therefore, we were not able to perform our experiments in absence of FCS which also contains ECM components as the reviewer rightly points out. We also did not use any protocols for surface occlusion, but Hakanpää et al. in their recent study on the influence of a wide range of ECM components on the formation of flat clathrin lattices (Hakanpää et al, 2023; doi.org/10.1083/jcb.202303107) point out that there likely is a kind of FCS-inherent surface occlusion mechanism. Since they also cultured their cells in the presence of serum, they were also wondering why ECM components from the FCS were not masking effects of the different ECM coatings they tested. Following up on this question, they showed, taking fibronectin as an example, that the amount of fibronectin that is deposited on glass coverslips after incubation in 10-100% FCS is very low "in line with similar experiments performed 30 years ago (Steele et al., 1992; doi.org/10.1002/jbm.820260704)". They hypothesize that this "occurs due to the high concentrations of BSA in serum (40 mg/ml) which

rapidly saturates the surface of culture dishes, thereby acting as a blocking agent for the binding of serum fibronectin".

This fits to the fact that even though FCS was added to all our coating conditions, we still observed striking differences in the formation of large non-canonical adhesions depending on the coating. Matrigel clearly suppressed their formation, while fibrillar collagen I promoted it. Since the coating is the only variable in these experiments (e.g. in Fig. S4), we hope that we could convince the reviewer that this is the decisive factor for the formation of large non-canonical adhesions in our experimental context, even though FCS components likely additionally influence the biogenesis process on a permissive substrate.

2) Figure 5 explores the dynamics of interconversion from FAs to RAs. This is an important finding (which incidentally comports with observations of bi-directional inter-convertibility not published with the original RA NCB paper). It is noted herein that $\beta 5$ levels remain largely unchanged during the transition from paxillin-positive to stonin1-positive. It would be interesting to assess $\beta 5$ integrin turnover kinetics around this transition (or, perhaps more feasibly, in adjacent adhesion structures that are paxillin- versus stonin1-positive). Given the reported differences in turnover rate / extent, it would be interesting to see if this difference aligns closely to the molecular markers used herein (paxillin vs stonin). Differences in integrin dynamics may be a fundamental 'biomarker' for differences in adhesion complex structure/function.

We were happy to learn that our findings of bi-directional inter-convertibility of FAs and RAs are confirmed by unpublished previous observations in an independent laboratory. We agree that it is very interesting to also study $\beta 5$ integrin turnover in paxillin-positive versus stonin1-positive adhesion structures. Since we have meanwhile generated a genome-edited cell line where not only stonin1, but also integrin $\beta 5$ is endogenously tagged, we used this cell line to preclude any influence on the turnover kinetics by integrin $\beta 5$ overexpression. In line with our view that integrin $\alpha V\beta 5$ constitutes the stable backbone on which other components dynamically exchange during the interconversion of FAs and RAs, we observed a very low fluorescence recovery after photobleaching (FRAP) for endogenously tagged integrin $\beta 5$ -mScarlet3, irrespective of whether it was residing in FAs or non-canonical adhesions. This suggests a very large immobile fraction of the molecule consistent with providing a stable scaffold structure. In comparison, EGFP-stonin1 as an "exchangeable component" displayed a much higher fluorescence recovery and thus a higher mobile pool within non-canonical adhesions.

These new data are now contained in Fig. 7, and the microscopy part of the methods section was expanded to include FRAP.

Surprised by the very low turnover kinetics of endogenously tagged integrin $\beta 5$, we performed as control the analogous experiment with overexpressed integrin $\beta 5$ as done in previous publications (e.g. in Lock et al.; doi.org/10.1038/s41556-018-0220-2). In fact, in line with Lock et al., we observed a higher exchange rate in non-canonical adhesions when using overexpressed integrin $\beta 5$ suggesting that the overexpression influences its turnover kinetics by supplying a larger pool of molecules for the recovery phase.

We provide this control experiment here as figure for the Reviewer:

Figure for the Reviewer:

Comparison of the fluorescence recovery of overexpressed integrin $\beta 5$ and endogenously labeled integrin $\beta 5$. C2C12 cells expressing endogenously tagged EGFP-stonin1 were either transfected with integrin $\beta 5$ -iRFP or genome-edited to express endogenously labeled integrin $\beta 5$ -mScarlet3. Two areas were chosen for photobleaching: one area within a stonin1-positive/ $\beta 5$ -positive large adhesion network and one area comprising stonin1-negative/ $\beta 5$ -positive focal adhesions. The comparison of the fluorescence recovery of overexpressed integrin $\beta 5$ and endogenously tagged integrin $\beta 5$ (see also main figure 7) revealed a lower extent of fluorescence recovery for endogenously labeled integrin $\beta 5$.

3) There is some occluded text on the right side of Fig 5 Panel D.

Unfortunately, we cannot spot this error in our manuscript version and assume that it might have been a file conversion artefact.

4) The observation of paxillin recruitment to existing non-canonical adhesions (e.g. retraction fibres), described in Figure 6, is interesting, and also comports with observations made (but not published) with the original RA NCB paper (and similarly, the formation (nucleation?) of small canonical adhesion punctae on the edge of / immediately adjacent to existing reticular adhesions, as shown clearly in Fig 6G). However, can the authors clarify in text when / if they are referring to a build up of e.g. paxillin, on non-canonical adhesions derived from a different cell (at any point in this section). While rapid build up of canonical components on non-canonical (e.g. retraction fibre) adhesions from the same cell is easy to understand, how 'orphan' integrins (and membrane?) attached to a surface (and external to the cell in question) can be used as a scaffold by cytoplasmic proteins, would seem to require some explanation. Do surface-residual membranes fuse with a passing cell, making the integrins in the residual membranes part of / accessible to the passing cell (and cytoplasmic proteins within that passing cell)? How might this otherwise be possible? This should at least be discussed, since it seems to imply some additional biophysical processes that are not yet considered herein. Unless I have missed something? Perhaps the emerging literature on migrasomes left as 'cues' by cells in 2D / 3D culture may provide some insights?

Again, we were happy to learn that our results are backed up by unpublished observations in other laboratories. Regarding the question whether paxillin is recruited to an integrin scaffold left behind by the same cell or by an independent cell, we now clarify in our manuscript: "upon respreading of a migratory cell across a $\beta 5$ integrin scaffold, which this cell had previously left behind as part of migratory retraction fibers, we observed the deposition of the focal adhesion marker paxillin onto the pre-existing paxillin-negative $\beta 5$ integrin structure". The migratory

retraction fibers which we imaged persistently maintained membrane connectivity with the cell body through the presence of tubular structures containing actin. As the reviewer points out, this is in fact the easier to understand scenario, as paxillin and integrin are throughout located within the same cell.

Reviewer #2:

In this manuscript entitled "Canonical and non-canonical integrin-based adhesions dynamically interconvert", Lukas et al. identify a protein marker called stonin1, which plays a role in the interconversion of canonical and non-canonical adhesions in conjugation with $\alpha V\beta 5$ integrin. The authors find that the stonin1- $\alpha V\beta 5$ integrin complex acts as an intermediate and decides the fate of the formation of canonical adhesions and non-canonical adhesions. The authors propose that non-canonical adhesive structures act as points of origin for generating canonical focal adhesions.

Although this work focuses on an important topic and overall, the experimental design/analysis is of good quality, my enthusiasm is diminished by the fact that the message describing the conversion of plaques into focal adhesions has been already published in Zuidema et al., 2022 J Cell Sci while two additional preprints show it as well (Bucher et al., bioarxiv doi: <https://doi.org/10.1101/493114>; and in Hakanpää et al., bioarxiv doi: <https://doi.org/10.1101/2022.05.30.494022>).

We thank the reviewer for highlighting the importance of our topic and the quality of our experimental design and analysis and hope to excite his/her enthusiasm for our manuscript further by providing in the following paragraph a detailed explanation in how far our finding of a conversion of non-canonical adhesions/plaques into focal adhesions is distinct from the great contributions the cited papers made to the field.

a) Zuidema et al. (2022 J Cell Sci; <https://doi.org/10.1242/jcs.259465>) show that integrin $\alpha V\beta 5$ can reside in focal adhesions as well as in flat clathrin lattices in line with our data and other previous data. Their paper focuses on the determinants of $\alpha V\beta 5$ localization within flat clathrin lattices and shows that the $\beta 5$ cytoplasmic domain binds more strongly to the endocytic adaptors ARH and Numb than to the focal adhesion components talin and kindlin. Besides, they dissect that phosphorylation of serines S759 and S762 within the SERS motif in the $\beta 5$ tail promotes $\beta 5$ localization within flat clathrin lattices. In addition, they demonstrate that "destabilization of microtubules promotes relocalization of integrin $\beta 5$ from FCLs into FAs through modulation of tension". The chosen phrasing of relocalization indicates that the authors envision the $\beta 5$ integrin to leave/disassemble the FCL structure and subsequently to join/assemble a FA. In fact, they do neither suggest nor present any evidence for a conversion mechanism where a pre-existing integrin $\alpha V\beta 5$ scaffold remains stable while the associated components exchange from clathrin plaque proteins to focal adhesion proteins which we show in our study.

b) Bucher et al. (bioarxiv doi: <https://doi.org/10.1101/493114>) study in their not-yet peer-reviewed preprint the origin of clathrin plaques. They show that FA-dependent ECM degradation serves as a topographical clue of unclear nature to induce clathrin plaques at the location of former FAs. In that way disassembling FAs, which leave behind integrins, are replaced by clathrin plaques in a process which they term "switch from FAs to clathrin plaques" and which is reminiscent of the conversion process from FAs to non-conventional adhesions which we describe. However, they do not study and do not show any evidence for the reverse process of a conversion of non-canonical adhesions/plaques into FAs.

c) Hakanpää et al. (preprint:doi.org/10.1101/2022.05.30.494022) analyze in their study (which has meanwhile been published in Journal of Cell Biology (<https://doi.org/10.1083/jcb.202303107>)) the role of ECM components and $\alpha 5\beta 1$ integrins for

the assembly of reticular adhesions and the exact relationship between reticular adhesions and flat clathrin lattices. Like the two other studies they do not propose or investigate any conversion from non-canonical adhesions into focal adhesions.

Thus, to the best of our knowledge, the conversion of non-canonical adhesions into focal adhesions has not been described in any previous publication.

Some conclusions drawn such as the exclusion of stonin1 and $\beta 5$ integrin from plaques are sometimes too generalized, lack molecular biochemical mechanisms, and are not controlled enough. For example, the present study lacks mechanistic insight into how modifications of $\alpha V\beta 5$ integrin affect the affinity and binding interactions with stonin1 and favor the formation of different adhesive structures.

To clarify, we do not conclude that stonin1 and $\beta 5$ integrin are excluded from plaques, but rather show in Fig. 1 that stonin1 and integrin $\alpha V\beta 5$ are present in different types of non-canonical adhesions including plaques.

Regarding the lack of controls, we have now added a large array of additional controls (including a new genome-edited cell line expressing endogenously tagged integrin $\beta 5$) as described in detail below in response to the individual comments of the reviewer.

Regarding mechanistic insights into how modifications of $\alpha V\beta 5$ integrin favor the formation of different adhesive structures, we now show that the findings of Zuidema et al. (<https://doi.org/10.1242/jcs.259465>) about integrin $\beta 5$ serine phosphorylation favoring its localization within flat clathrin lattices also holds true for the larger adhesion networks we describe (Fig. S9 B-C), see also our comments further below.

Also, the entire paper focuses on undifferentiated C2C12 myoblasts (this cell line is known to be a poor model for clathrin plaque formation) while plaques have been extensively characterized in differentiated myotubes as their number sharply increases after differentiation.

While it is true that Moulay et al. (doi.org/10.1083/jcb.201912061) show a larger number of clathrin plaques in myotubes as compared to myoblasts, they also stress that clathrin plaques are present in myoblasts. Unfortunately, the culture conditions for myoblasts are not entirely clear in this paper, but for some Matrigel is mentioned as coating. We show that Matrigel suppresses the formation of non-conventional adhesions, while other conditions such as collagen I coating clearly promote their biogenesis. Therefore, the ability of C2C12 cells to form plaques depends largely on the coating on which they are grown.

To confirm our results in independent cell types, we had already included experiments with primary fibroblasts, primary myoblasts and primary astrocytes in Fig. S2G of our submitted manuscript (now Fig. S2H) demonstrating that all these primary cells are able to form large stonin1-positive networks. Following the reviewer's suggestion we now extended this collection to an additional cell line, demonstrating that our findings can also be recapitulated in the human retinal pigment epithelium cell line hTERT-RPE1. These new results were added to supplementary Fig. S2 as new panel S2G.

Major comments:

1/ One of the major claims concerns the successive transition from focal adhesions to RAs/plaques without altering the integrin scaffold. Although this observation is interesting, $\alpha V\beta 5$ localization pattern is regulated by the amount of cellular tension, it has already been shown by the Sonnenberg group (Zuidema et al., 2018) that when tension is increased, integrin $\alpha V\beta 5$ is localized predominantly in FAs and switch to FCLs when cellular tension is reduced. Here the authors have not taken into consideration how cellular tension affects the stonin1

localization in conjugation with $\alpha V\beta 5$, while this might seem trivial, in the particular work it is a major issue, as it defines the interplay between canonical and non-canonical adhesions.

Indeed, the Sonnenberg lab shows in Zuidema et al. 2018 and 2022 that cellular tension is one determinant which influences $\alpha V\beta 5$ localization, with low tension favoring the presence of $\alpha V\beta 5$ integrins in non-canonical adhesions. They show the effect of cellular tension mainly via strong manipulations: To increase tension, they employ overexpression of a constitutively active RhoA mutant. To abrogate tension, they use either overexpression of a constitutively inactive mutant of RhoA or the myosin inhibitor blebbistatin. In line with their data and data from the Humphries lab (Lock et al., doi.org/10.1038/s41556-018-0220-2) on reticular adhesions, we show in our Fig. 6G and S9A that blebbistatin treatment and thus low tension is compatible with the maintenance of large stonin1-positive non-canonical adhesion networks.

When the Sonnenberg lab analyzed the steady-state tension level of different cell lines by immunoblotting for phosphorylated myosin light chain (pMLC), they observed a certain degree of correlation between high pMLC levels and $\beta 5$ integrin residence in FAs, but stated also "when all the colon and breast cancer cell lines were taken together, this relationship was less clear".

To test the relevance of tension under our coating conditions, we now assessed tension like the Sonnenberg group by immunoblotting for pMLC (Fig. S4E). However, we did not observe a difference in pMLC levels between cells grown on collagen I which promotes non-canonical adhesion formation and cells grown on Matrigel which suppresses non-canonical adhesion biogenesis. Therefore, different levels of tension do not appear to play a major role for the formation of non-canonical adhesions under our cell culture conditions. Instead the type of coating appears to be decisive in line with the recent publication by Hakanpää et al. (https://doi.org/10.1083/jcb.202303107). Matrigel largely contains laminin which Hakanpää et al. showed to lead to the deposition of fibronectin which in turn prevents the formation of flat clathrin lattices and reticular adhesions via active $\alpha 5\beta 1$ integrin, while collagen I does not have this inhibitory effect.

The papers from the Sonnenberg group also point out that tension is not the only determinant, but phosphorylation of the $\beta 5$ integrin SERS motif is also important for the presence of $\beta 5$ integrin in non-canonical adhesions.

Therefore, as mentioned earlier, we tested the influence of serine phosphorylation on $\beta 5$ integrin localization in large non-canonical adhesion networks. For that, we expressed phospho-dead (S759A/S762A) or phospho-mimetic (S759E/S762E) $\beta 5$ integrin mutants. We find that the phospho-mimetic version of $\beta 5$ resides to a larger degree in non-canonical adhesion networks (new supplementary Fig. S9B-C), as was previously shown for RAs/plaques by Zuidema et al. (2022 J Cell Sci; https://doi:10.1242/jcs.259465).

2/The PREM correlative imaging in Fig. 4 is a strong argument in favour of the fact that stonin1 and integrin beta5 are not exclusively localized in clathrin lattices. However, the authors should perform additional controls. For instance, $\beta 5$ is always overexpressed and could be deleterious to the localization of clathrin or other components. The authors should at least analyse the distribution of endogenous $\beta 5$ integrin using antibodies instead of overexpressing it stably.

We now have monitored in several ways that the overexpressed $\beta 5$ integrin which we use localizes and behaves the same way as $\beta 5$ integrin which is present at endogenous levels.

Previously, we had already immunolabeled the different endogenously occurring non-canonical adhesions with an αV integrin-specific antibody and demonstrated its colocalization with stonin1 (Fig. 1B-D). Unfortunately, we are not aware of a functional $\beta 5$ integrin-specific antibody which recognizes murine $\beta 5$ and thus could be used in C2C12 cells. However, since

we now also recapitulate our findings in hTERT-RPE1 cells, we used those to confirm a precise co-localization between immunolabeled endogenous integrin $\beta 5$ and integrin αV in non-canonical adhesions. Furthermore, the examination of stonin1 localization in conjunction with integrin αV demonstrated a perfect overlap, underscoring the close relationship between stonin1 and endogenous integrin αV and $\beta 5$ within adhesion networks, while clathrin exhibited also in this experiment a more punctate pattern (now presented in Fig. S2G).

In addition, we also successfully generated a genome-edited cell line expressing endogenous $\beta 5$ integrin c-terminally tagged with the red fluorescent protein mScarlet3. We demonstrate that the large adhesion networks formed in these cells have the same appearance as the ones formed in stably $\beta 5$ integrin overexpressing cells and show the same type of clathrin distribution ruling out that the overexpressed $\beta 5$ integrin is deleterious to clathrin localization (presented in Fig. S6).

3/The authors show regions where Stonin1 and $\beta 5$ integrin colocalize but no clathrin is present. However, the EM images are not clearly showing what is below the fluorescent signal. Are these clumps of overexpressed $\beta 5$? Are there any vesicles? This is an important point considering the fact that previous work has shown localization of $\beta 5$ in clathrin lattices and RAs. Have the CLEM data been validated with immunogold labeling the endogenous proteins?

We thank the reviewer for this valid concern. To show what is below the fluorescent signal, we provide the platinum replica EM images as single channels with a zoom into the relevant region in Fig. 4E,F. These images depict plasma membrane sheets with detailed insights of structural components present at the cytosolic plasma membrane leaflet. During generation of plasma membrane sheets all cytosolic components not bound to the plasma membrane (such as vesicles) are removed. Clathrin, actin or caveolae are easy to identify because of their unique structural characteristics (see explanatory arrows in PREM image below).

The "clumps" mentioned by the reviewer show protein complexes located at or within the plasma membrane and are also present on PREM images of cells which do not overexpress any protein. The correlative fluorescence light image clearly indicates that in this region (marked by yellow boxes in images) integrin $\beta 5$ (magenta) and stonin1 (cyan) colocalize. We agree with the reviewer that the CLEM approach cannot determine if any other proteins are also located in this region. However, currently no (imaging) method is able to inspect plasma membrane regions in high resolution and determine all of its associated proteins.

Therefore, we strongly believe that our CLEM approach is suitable to (1) determine clathrin-negative adhesions and (2) precisely correlate and therefore identify two major components of these adhesions - integrin $\beta 5$ and stonin1. We did not use immunogold labeling in these CLEM experiments, since no EM-suitable antibodies were available for murine integrin $\beta 5$, and immunogold labeling is a difficult technique which can result in large amounts of background

gold staining making it very difficult to evaluate the obtained results without adequate knockout controls.

However, as we were discussing in response to question 2, we did not see any differences in integrin $\beta 5$ localization in cells overexpressing fluorescently labeled integrin $\beta 5$ (Fig. 3C-F; 5B-C; 6B-D) and cells either labeled with human integrin $\beta 5$ -specific antibodies (Fig. S2G) or expressing endogenously labeled integrin $\beta 5$ (Fig. S6, S7, S10, S11).

4/ On line 202 the authors state “First, our experiments in migrating cells confirmed that RAs/plaques can form de novo, although such events appeared to be rare (Fig. S5A-C) “. Here in this context, the authors need to provide kymographs of clathrin assembly in comparison with stonin1. Here the authors have not taken into consideration endogenous $\alpha V\beta 5$ integrin contributions in these RAs/Plaques. Quantification of these events would be important before generalizing the phenomena.

Using our newly generated cell line endogenously expressing EGFP-stonin1 and integrin $\beta 5$ -mScarlet3 and transduced with fluorescently-tagged clathrin, we now imaged additional examples of the rare de novo formation of non-canonical adhesions and show that integrin $\beta 5$, stonin1 and clathrin appear simultaneously during this process. We now depict this data as part of supplementary Fig. S7 together with a quantification.

5/ The present study focuses on endogenously tagged EGFP-stonin1, but stably overexpressed $\beta 5$ -integrin throughout the study. Expression levels of these proteins play a role in the stonin1- $\alpha V\beta 5$ integrin scaffold. The authors need to provide endogenously tagged $\beta 5$ -integrin that doesn't affect the expression levels of all the mentioned canonical and non-canonical markers. The authors should control this by showing that the tagged and untagged stonin 1 level doesn't affect clathrin, and actin expression levels by using western blot.

Following the reviewer's suggestion, we now indeed went to the effort of creating a genome-edited cell line where integrin $\beta 5$ is endogenously tagged with mScarlet3. As suggested by the reviewer we show in the new Fig. S3 that neither our previous stable overexpression of integrin $\beta 5$ -iRFP nor the endogenous tagging of $\beta 5$ with mScarlet3 or the endogenous tagging of stonin1 with EGFP has any adverse effect on the protein level of either the canonical FA marker paxillin or the endocytic protein clathrin or actin as quantified by western blot.

6/ The study focuses exclusively on C2C12 cells, but what about other cell types, formation of canonical and non-canonical adhesions differ in various cell types. It is very important to test stonin1 behavior in other cell types before considering the novel marker for both canonical and Reticular adhesions/clathrin plaques. Have the authors tried using Hela cells which also form plaques?

It is of course a very valid point that cell biological findings should be replicable in independent cell lines and preferentially also primary cells. Therefore, we present in our Fig. S2H a panel showing the formation of the large stonin1-positive networks in primary myoblasts, primary astrocytes and primary fibroblasts. Furthermore, we have extended this analysis now to hTERT-RPE1 human retinal pigment epithelial cells. We chose this cell line over HeLa cells since in contrast to the tumor-derived HeLa cells hTERT-RPE1 cells are non cancerous, since they have been immortalized via human telomerase reverse transcriptase (hTERT), thus combining features of primary cells with the proliferative capacity of a cell line. These new results can be found in supplementary Fig. S2G.

7/ The present study focuses on RAs/plaques that can also arise during focal adhesion disassembly by recycling the existing $\alpha V\beta 5$ scaffold. It is very important to test this for instance by using FRAP experiments to show the same $\alpha V\beta 5$ facilitates interacting with stonin1 and canonical protein markers before generalizing the proposed model scenario.

This is a very good suggestion that was also made by reviewer 1 and as we discussed already in answer to his/her comment: We agree that it is very interesting to also study $\beta 5$ integrin turnover in paxillin-positive versus stonin1-positive adhesion structures. We used our newly generated doubly genome-edited cell line for this experiment to preclude any influence on the turnover kinetics by integrin $\beta 5$ overexpression. In line with our view that integrin $\alpha V\beta 5$ constitutes the stable backbone on which other components dynamically exchange during the interconversion of FAs and RAs, we observed a very low fluorescence recovery after photobleaching (FRAP) for endogenously tagged integrin $\beta 5$ -mScarlet3, irrespective of whether it was residing in focal adhesions or non-canonical adhesions. This suggests a very large immobile fraction of the molecule consistent with providing a long-term scaffold structure. In comparison, EGFP-stonin1 as an "exchangeable component" displayed a much higher fluorescence recovery and thus a higher mobile pool within non-canonical adhesions. These new data are now contained in Fig. 7.

8/ On line 253 the authors state that "However, in presence of non-canonical adhesion networks (i.e. on collagen), small focal adhesions appeared also centrally right next to the networks (Fig. 6G, H) suggesting again a novel mode of focal adhesion assembly that rather depends on pre-existing $\alpha V\beta 5$ integrin scaffolds than on lamellipodial protrusion". The authors need to provide CLEM ultra-structural views on such conditions i.e. synchronized focal adhesion assembly events adjacent to non-canonical adhesive structures.

While CLEM is a great tool to visualize clathrin lattices, we did not opt for this method in this context since it is limited to static snap-shots and cannot easily be used to study dynamic processes. We fully agree that there are many mechanistic questions which still have to be elucidated in the future regarding this mode of focal adhesion assembly. Therefore, we now added an additional sentence in the revised manuscript to stress the point that "**molecular details behind this mechanism remain elusive at the moment**".

9/ The present study focuses on endogenously tagged EGFP-stonin1, but overexpression of beta5 integrin which could perturb the clathrin homeostasis and therefore contribute to the difference between this work and previously published papers on plaques showing the presence of endogenous $\beta 5$ integrin in clathrin plaques.

We agree that overexpression artefacts are always a valid concern. Therefore, we followed the earlier suggestion of the reviewer to generate a cell line where integrin $\beta 5$ is endogenously tagged. As discussed in reply to point 5, we also verified that clathrin levels are not perturbed in this and the other cell lines we used (Fig. S3). With the new genome-edited integrin $\beta 5$ cell line we could confirm the presence of endogenous $\beta 5$ integrin in clathrin plaques (Fig. S10 D-G) and in large non-canonical adhesion networks (Fig. S6) in agreement with previous papers showing the localization of $\beta 5$ integrin in clathrin plaques.

10/ On line 636 the authors state "Percentage of stonin1-positive adhesion network area that is positive for clathrin (N=4 unroofed cells)". The number of cells but not the number of independent replicates and the number of clathrin structures considered for the analysis. One

should know statistics on these structures, and also how big are these stonin1 and clathrin-positive structures. These statistics give a morphometric analysis of these ultra-structures before coming to conclusion. Have these experiments in Figure 4 and Fig S4 been performed only once?

We apologize for this omission. We now added also the number of independent experiments as well as the total number of clathrin structures detected during CLEM inspections to the legend of Fig. 4. For the CLEM analysis, we inspected in total 8 plasma membrane regions out of 4 cells in 2 independent CLEM experiments. We investigated 617 Stonin1 spots, of which 133 were positive for clathrin.

We also measured now the size of the stonin1-positive clathrin structures based on the well recognizable clathrin lattices and included it in Fig. 4 as panel D.

11/ Clathrin lattices were detected throughout the manuscript with antibodies against the clathrin light chain. Does this antibody recognize both light chains isoforms CLCa and CLCb? The authors should at least once validate the absence of clathrin in beta5/stonin1 regions with a clathrin heavy chain antibody which might also prove more reactive.

We used the clathrin light chain-specific antibody from Santa Cruz Biotechnology (sc-12735). This antibody is recommended for the detection CLCa und CLCb. As suggested, we have now validated our clathrin light chain antibody-based results with an antibody against clathrin heavy chain which indeed provides the same outcome as depicted in the new Fig. S2G and S6.

Minor comments:

-Typo in the sentence in line 58 “heterdimers, provide the molecular link between RAs/plaques and the extracellular matrix” needs to be changed to heterodimers.

We thank the reviewer for his/her careful reading and have now corrected this mistake.

The legends in a several panels (i.e. Figure 5D and Figure 6E) are not clear enough.

We thank the reviewer for this remark. We have now included additional information in the mentioned figure legends to make sure that the experimental analysis is clear to the reader.

We hope that we were able to address all remaining topics and look forward to the publication of our results.

Reviewer #1 (Remarks to the Author):

Overall my view is that this is a significantly improved manuscript, that makes clear contributions to illuminating the life cycles of non-canonical adhesion complexes, i.e. reticular adhesions / plaques, and their bi-directional inter-convertability with each other. I believe that this has important implications for the overall view of how adhesion complexes arise.

I am satisfied with the explanation regarding the use of FCS in culture of cells on previously (purified) ECM coatings. Indeed, I agree that the differences in observed outcomes (in terms of adhesion types) implicate the initial purified ECM as the cause. Perhaps in future, it may be worth repeating this without FCS but with purified growth factors that may permit reticular adhesion formation in a more 'ECM-controlled' context, but I regard this as non-critical and perhaps beyond the scope of this paper.

The updated FRAP experiments add to the notion of the $\alpha 5 \beta 1$ integrin as a backbone / scaffold that can initiate or sustain adhesion structures of both canonical and non-canonical forms. On this point, I would note that it is not entirely clear in the figure 7 legend how many individual FRAP locations were assessed in total, across how many cells, across the 3 independent experiments. Ideally, I feel this should be made clear in the legend. I realise that the number of bleach locations is noted (2 per adhesion type per cell) in methods.

1) I would recommend placing all of this information in a single location (ideally legend) so that the reader doesn't have to compile the numbers from multiple locations.

Regarding Figure 5D, I no longer see the text occlusion, so likely this was indeed a technical glitch.

Regarding the issue of re-spreading on previous adhesions. I believe I understand the Authors meaning, and appreciate their updated text, but I still feel that even with the updated text, the meaning could still be ambiguous to some readers. I think perhaps it is the words 'left behind' that contribute to this - since this might easily be interpreted as implying adhesions (and associated membrane) that were previously detached from the cell, as per migrasomes etc. This is certainly observed with reticular adhesions in retraction settings. It seems to me that the authors mean the cell has changed direction and is re-spreading over its own trailing-edge, so to speak, and thus 'trailing-edge' (reticular adhesion) retraction structures are converted back to 'leading-edge' canonical (focal) adhesion structures, as indicated by paxillin recruitment to the pre-existing structures. It is important (in my opinion) to emphasise that this happens BEFORE the trailing-edge adhesions are detached from the cell, since recruiting paxillin to adhesions that were fully detached from the cell would require some further explanation (or at least clear identification that this needs future explanation).

2) I believe the authors should try to further clarify this.

Reviewer #2 (Remarks to the Author):

In this revised version of the manuscript by Lukas et. al., entitled "Canonical and non-canonical integrin-based adhesions dynamically interconvert" the authors now provide several additional experiments, such as additional analysis on newly generated genome-edited cell line expressing endogenous $\beta 5$ integrin. These data are convincing, nicely presented, and strengthen the overall conclusions. In addition, the analysis of additional primary cells and the turnover kinetics of EGFP-stonin1 strengthen the observations. Although the authors have satisfactorily addressed my previous comments, I am still not convinced by the conclusions and the data presented in Figure S4 on membrane tension. Analyzing pMLC immunoblots is not sufficient to draw conclusions on how membrane tension affects the distribution of $\beta 5$ and stonin. The results showing an almost complete disappearance of the stonin $\beta 5$ signal in cells grown on matrigel is also puzzling. In our hands, matrigel was used to discover flat clathrin plaques in myotubes, and not only does not inhibit the formation of flat clathrin lattices, but on the contrary, it promotes the formation of a complex extracellular matrix, which in turn favors the formation of large flat clathrin lattices. Part of this discrepancy could stem from the fact that all the experiments were performed on undifferentiated myoblasts instead of differentiated myotubes, as it is the myotubes that produce

clathrin plaques.

Manuscript NCOMMS-23-09818 - Point-by-point response to the reviewer's comments

Reviewer #1

Overall my view is that this is a significantly improved manuscript, that makes clear contributions to illuminating the life cycles of non-canonical adhesion complexes, i.e. reticular adhesions / plaques, and their bi-directional inter-convertability with each other. I believe that this has important implications for the overall view of how adhesion complexes arise.

We thank the reviewer for his/her positive evaluation of the impact of our manuscript.

I am satisfied with the explanation regarding the use of FCS in culture of cells on previously (purified) ECM coatings. Indeed, I agree that the differences in observed outcomes (in terms of adhesion types) implicate the initial purified ECM as the cause. Perhaps in future, it may be worth repeating this without FCS but with purified growth factors that may permit reticular adhesion formation in a more 'ECM-controlled' context, but I regard this as non-critical and perhaps beyond the scope of this paper.

We agree that it will be important in the future to dissect which exact FCS components are critical for the growth of non-canonical adhesions.

The updated FRAP experiments add to the notion of the $\alpha 5 \beta 1$ integrin as a backbone / scaffold that can initiate or sustain adhesion structures of both canonical and non-canonical forms. On this point, I would note that it is not entirely clear in the figure 7 legend how many individual FRAP locations were assessed in total, across how many cells, across the 3 independent experiments. Ideally, I feel this should be made clear in the legend. I realise that the number of bleach locations is noted (2 per adhesion type per cell) in methods. 1) I would recommend placing all of this information in a single location (ideally legend) so that the reader doesn't have to compile the numbers from multiple locations.

Following the reviewer's suggestion, we have now added all this information to the legend of figure 7C.

Regarding Figure 5D, I no longer see the text occlusion, so likely this was indeed a technical glitch.

We are happy that this is resolved.

Regarding the issue of re-spreading on previous adhesions. I believe I understand the Authors meaning, and appreciate their updated text, but I still feel that even with the updated text, the meaning could still be ambiguous to some readers. I think perhaps it is the words 'left behind' that contribute to this - since this might easily be interpreted as implying adhesions (and associated membrane) that were previously detached from the cell, as per migrasomes etc. This is certainly observed with reticular adhesions in retraction settings. It seems to me that the authors mean the cell has changed direction and is re-spreading over its own trailing-edge, so to speak, and thus 'trailing-edge' (reticular adhesion) retraction structures are converted back to 'leading-edge' canonical (focal) adhesion structures, as indicated by Paxillin recruitment to the pre-existing structures. It is important (in my opinion) to emphasise that this happens BEFORE the trailing-edge adhesions are detached from the cell, since recruiting Paxillin to adhesions that were fully detached from the cell would require some further explanation (or at least clear identification that this needs future explanation).

The reviewer correctly explains what we wanted to convey in our manuscript.

2) I believe the authors should try to further clarify this.

We have now further clarified this point in our manuscript by describing a " β 5 integrin scaffold located within a migratory retraction fiber which the cell previously formed and which is still connected to its cell body".

Reviewer #2

In this revised version of the manuscript by Lukas et. al., entitled "Canonical and non-canonical integrin-based adhesions dynamically interconvert" the authors now provide several additional experiments, such as additional analysis on newly generated genome-edited cell line expressing endogenous β 5 integrin. These data are convincing, nicely presented, and strengthen the overall conclusions. In addition, the analysis of additional primary cells and the turnover kinetics of EGFP-stonin1 strengthen the observations.

We are happy that the reviewer agrees that our additional experiments are convincing and lend further support to our conclusions.

Although the authors have satisfactorily addressed my previous comments, I am still not convinced by the conclusions and the data presented in Figure S4 on membrane tension. Analyzing pMLC immunoblots is not sufficient to draw conclusions on how membrane tension affects the distribution of beta5 and stonin.

We agree that the analysis of pMLC immunoblots can only be a first step towards dissecting the impact of membrane tension since it lacks for instance spatial resolution. However, since this was not the focus of our manuscript, we did not go into further detail here but rather chose to provide additional evidence for the main conclusion of our paper, the existence of a conversion mechanism between canonical and non-canonical adhesions.

The results showing an almost complete disappearance of the stonin beta 5 signal in cells grown on matrigel is also puzzling. In our hands, matrigel was used to discover flat clathrin plaques in myotubes, and not only does not inhibit the formation of flat clathrin lattices, but on the contrary, it promotes the formation of a complex extracellular matrix, which in turn favors the formation of large flat clathrin lattices. Part of this discrepancy could stem from the fact that all the experiments were performed on undifferentiated myoblasts instead of differentiated myotubes, as it is the myotubes that produce clathrin plaques.

We also think that this discrepancy might arise from the fact that we mostly used myoblasts instead of differentiated myotubes. We would like to propose that the difference in culture length might be the decisive factor here since we and others have shown that long-term culture strongly promotes non-canonical adhesions. For the experiment in question (Fig. 2c & S4a,b) C2C12 myoblasts were cultured for just 24 h on Matrigel or collagen (an important information we now added to the figure legend). Under these circumstances collagen promotes and Matrigel suppresses non-canonical adhesion formation. However, if we leave C2C12 cells in culture for 4-10 days (similar to Fig. S2b-d), they will form adhesions independently of the initial coating. We assume that this is due to the fact that the surface Matrigel coating will be degraded by proteolysis over time.

Since the protocol to obtain myotubes requires a several days long differentiation step after the initial cell seeding on Matrigel, the Matrigel matrix that was originally supplied might well be degraded over time at the sites of the observed non-canonical adhesions. Alternatively,

the non-canonical adhesion promoting effects of long-term culture could override the inhibitory influences of Matrigel that we observe in short-term culture in a different manner. Clearly, additional experiments will be necessary in the future to fully understand all factors that influence non-canonical adhesion formation to unambiguously resolve this point.